# Spectral Invariant Learning for Dynamic Graphs under Distribution Shifts

**Zeyang Zhang[1]\***, **Xin Wang[1]†**, **Ziwei Zhang[1]**, **Zhou Qin[2]**,
**Weigao Wen[2]**, **Hui Xue[2]**, **Haoyang Li[1]**, **Wenwu Zhu[1]†**

[1]Department of Computer Science and Technology, BNRist, Tsinghua University, [2]Alibaba Group

zy-zhang20@mails.tsinghua.edu.cn, {xin_wang, zwzhang}@tsinghua.edu.cn,
{qinzhou.qinzhou, weigao.wen, hui.xueh}@alibaba-inc.com,
lihy18@mails.tsinghua.edu.cn, wwzhu@tsinghua.edu.cn

## Abstract

Dynamic graph neural networks (DyGNNs) currently struggle with handling distribution shifts that are inherent in dynamic graphs. Existing work on DyGNNs with out-of-distribution settings only focuses on the time domain, failing to handle cases involving distribution shifts in the spectral domain. In this paper, we discover that there exist cases with distribution shifts unobservable in the time domain while observable in the spectral domain, and propose to study distribution shifts on dynamic graphs in the spectral domain for the first time. However, this investigation poses two key challenges: i) it is non-trivial to capture different graph patterns that are driven by various frequency components entangled in the spectral domain; and ii) it remains unclear how to handle distribution shifts with the discovered spectral patterns. To address these challenges, we propose Spectral Invariant Learning for Dynamic Graphs under Distribution Shifts (**SILD**), which can handle distribution shifts on dynamic graphs by capturing and utilizing invariant and variant spectral patterns. Specifically, we first design a DyGNN with Fourier transform to obtain the ego-graph trajectory spectrums, allowing the mixed dynamic graph patterns to be transformed into separate frequency components. We then develop a disentangled spectrum mask to filter graph dynamics from various frequency components and discover the invariant and variant spectral patterns. Finally, we propose invariant spectral filtering, which encourages the model to rely on invariant patterns for generalization under distribution shifts. Experimental results on synthetic and real-world dynamic graph datasets demonstrate the superiority of our method for both node classification and link prediction tasks under distribution shifts.

## 1 Introduction

Dynamic graph neural networks (DyGNNs) have achieved remarkable success in many predictive tasks over dynamic graphs [1, 2]. Existing DyGNNs exhibit limitations in handling distribution shifts, which naturally exist in dynamic graphs due to multiple uncontrollable factors [3, 4, 5, 6]. Existing work on out-of-distribution generalized DyGNNs focuses on handling distribution shifts in the time domain. For example, DIDA [7] utilizes dynamic graph attention to mask the graph trajectories to capture the invariant patterns on dynamic graphs, which assumes that in the time domain, the distribution shift is observable and the invariant and variant patterns can be easily disentangled.

However, there exist cases that the distribution shift is unobservable in the time domain while observable in the spectral domain, as shown in Figure 1. The shift in frequency components can

---

\*This work was done during the author's internship at Alibaba Group

†Corresponding authors

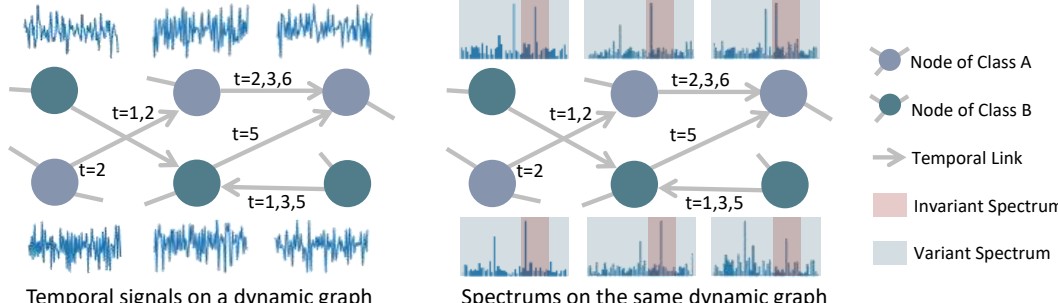

Figure 1: An illustration example: the graph dynamics from different frequency components are entangled in the temporal domain, while it is much easier to distinguish different frequency components by masking the spectrums in the spectral domain. In this case, the frequency components in the invariant spectrums determine the node labels, while the relationship between the variant spectrums and labels is not stable under distribution shifts.

be clearly observed in the spectral domain, while these components are indistinguishable in the time domain. Moreover, in real-world applications, the observed dynamic graphs usually consist of multiple mixed graph structural and featural dynamics from various frequency components [8, 9, 10].

To address this problem, in this paper, we study the problem of handling distribution shifts on dynamic graphs in the spectral domain for the first time, which poses the following two key challenges: i) it is non-trivial to capture different graph patterns that are driven by various frequency components entangled in the spectral domain, and ii) it remains unclear how to handle distribution shifts with the discovered spectral patterns.

To tackle these challenges, we propose Spectral Invariant Learning for Dynamic Graphs under Distribution Shifts (**SILD**[3]). Our proposed **SILD** model can effectively handle distribution shifts on dynamic graphs by discovering and utilizing invariant and variant spectral patterns. Specifically, we first design a DyGNN with Fourier transform to obtain the ego-graph trajectory spectrums so that the mixed graph dynamics can be transformed into separate frequency components. Then we develop a disentangled spectrum mask that leverages the amplitude and phase information of the ego-graph trajectory spectrums to obtain invariant and variant spectrum masks so that graph dynamics from various frequency components can be filtered. Finally, we propose an invariant spectral filtering that discovers the invariant and variant patterns via the disentangled spectrum masks, and minimize the variance of predictions with exposure to various variant patterns. As such, **SILD** is able to exploit invariant patterns to make predictions under distribution shifts. Experimental results on several synthetic and real-world datasets, including both node classification and link prediction tasks, demonstrate the superior performance of our **SILD** model compared to state-of-the-art baselines under distribution shifts. To summarize, we make the following contributions:

- We propose to study distribution shifts on dynamic graphs in the spectral domain, to the best of our knowledge, for the first time.

- We propose Spectral Invariant Learning for Dynamic Graphs under Distribution Shifts (**SILD**), which can handle distribution shifts on dynamic graphs in the spectral domain.

- We employ DyGNN with Fourier transform to obtain the node spectrums, design a disentangled spectrum mask to obtain invariant and variant spectrum masks in the spectral domain, and propose the invariant spectral filtering mechanism so that **SILD** is able to handle distribution shifts.

- We conduct extensive experiments on several synthetic and real-world datasets, including both node classification and link prediction tasks, to demonstrate the superior performance of our method compared to state-of-the-art baselines under distribution shifts.

---

[3]The codes are available at Github.

## 2 Problem Formulation and Notations

**Dynamic Graphs** A dynamic graph can be represented as $\mathcal{G} = (\{\mathcal{G}^t\}_{t=1}^T)$, where $T$ represents the total number of time stamps, and each $\mathcal{G}^t = (\mathcal{V}^t, \mathcal{E}^t)$ corresponds to a graph snapshot at time stamp $t$ with the node set $\mathcal{V}^t$ and the edge set $\mathcal{E}^t$. For simplicity, we also represent a graph snapshot as $\mathcal{G}^t = (\mathbf{X}^t, \mathbf{A}^t)$, which includes the node feature matrix $\mathbf{X}^t$ and the adjacency matrix $\mathbf{A}^t$. We further denote a random variable of $\mathcal{G}^t$ as $\mathbf{G}^t$. The prediction task on dynamic graphs aims to utilize past graph snapshots to make predictions, *i.e.*, $p(\mathbf{Y}^t | \mathbf{G}^{1:t})$, where $\mathbf{G}^{1:t} = \{\mathbf{G}^1, \mathbf{G}^2, \ldots, \mathbf{G}^t\}$ denotes the graph trajectory, and the label $\mathbf{Y}^t$ represent the node properties or the links at time $t + 1$. For brevity, we take node-level prediction tasks as an example in this paper. Following [7], the distribution of graph trajectory can be factorized into ego-graph trajectories, such that $p(\mathbf{Y}^t | \mathbf{G}^{1:t}) = \prod_v p(\mathbf{y}_v^t | \mathbf{G}_v^{1:t})$.

**Distribution Shifts on Dynamic Graphs** The common optimization objective for prediction tasks on dynamic graphs is to learn an optimal predictor with empirical risk minimization (ERM), *i.e.* $\min_\theta \mathbb{E}_{(y^t, \mathcal{G}_v^{1:t}) \sim p_{tr}(\mathbf{y}^t, \mathbf{G}_v^{1:t})} \mathcal{L}(f_\theta(\mathcal{G}_v^{1:t}), y^t)$, where $f_\theta$ is a learnable dynamic graph neural networks. Under distribution shifts, however, the optimal predictor trained with ERM and the training distribution may not generalize well to the test distribution, since the risk minimization objectives under two distributions are different due to $p_{tr}(\mathbf{Y}^t, \mathbf{G}^{1:t}) \neq p_{te}(\mathbf{Y}^t, \mathbf{G}^{1:t})$. The distribution shift on dynamic graphs is complex that may originate from temporal distribution shifts [11, 6, 12, 13, 14] as well as structural distribution shifts [15, 16, 17]. For example, trends or community structures can affect interaction patterns in co-author networks [18] and recommendation networks [19], *i.e.*, the distribution of ego-graph trajectories may vary through time and structures.

Following out-of-distribution (OOD) generalization literature [7, 15, 11, 20, 21, 22], we make the following assumptions of distribution shifts on dynamic graphs:

**Assumption 1** *For a given task, there exists a predictor $f(\cdot)$, for samples $(\mathcal{G}_v^{1:t}, y^t)$ from any distribution, there exists an invariant pattern $P_I^t(v)$ and a variant pattern $P_V^t(v)$ such that the following conditions are satisfied: 1) the invariant patterns are sufficient to predict the labels, $y_v^t = f(P_I^t(v)) + \epsilon$, where $\epsilon$ is a random noise, 2) the observed data is composed of invariant and variant patterns, $P_I^t(v) = \mathcal{G}_v^{1:t} \backslash P_V^t(v)$, 3) the influence of the variant patterns on labels is shielded by the invariant patterns, $\mathbf{y}_v^t \perp \mathbf{P}_V^t(v) \mid \mathbf{P}_I^t(v)$.*

In the next section, inspired by [16], we give a motivation example to provide some high-level intuition before going to our formal method.

## 3 Motivation Example

Here we introduce a toy dynamic graph example to motivate learning invariant patterns in the spectral domain. We assume that the invariant and variant patterns lie in the 1-hop neighbors, *i.e.*, each node has an invariant subgraph and a variant subgraph. For simplicity, we focus on the number of neighbors, *i.e.*, each node $v$ has an invariant subgraph related degree $\mathbf{d}_{v,1} \in \mathbb{R}^{T \times 1}$ and a variant subgraph related degree $\mathbf{d}_{v,2} \in \mathbb{R}^{T \times 1}$. Only the former determines the node label, *i.e.*, $y_v = \mathbf{g}^\top \mathbf{d}_{v,1}$. Note that invariant and variant subgraphs are not observed in the data. We further assume a one-dimensional constant feature for each node, which is set as 1 without loss of generality.

For the model in the spatial-temporal domain, we adopt sum pooling as one-layer graph convolution, *i.e.*, the message passing for each node and time is $\mathbf{h}_v = \sum_{u \in \mathcal{N}_v} 1 = \mathbf{d}_{v,1} + \mathbf{d}_{v,2}$. We further adopt a mask $\mathbf{m} \in \mathbb{R}^{T \times 1}$ to filter patterns in the temporal domain and make predictions by a linear classifier, *i.e.*, $\hat{y}_v = \mathbf{w}^\top (\mathbf{m} \odot \mathbf{h}_v)$, where $\mathbf{w} \in \mathbb{R}^{T \times 1}$ denotes the learnable parameters. Then, the empirical risk in the training dataset $D_{tr}$ is $R_{tr}(\mathbf{w}) = \frac{1}{|D_{tr}|} \sum_{v \in D_{tr}} (\hat{y}_v - y_v)^2$. We have the following proposition.

**Proposition 1** *For any mask $\mathbf{m} \in \mathbb{R}^{T \times 1}$, for the optimal classifier in the training data $\mathbf{w}^* = \arg\min_{\mathbf{w}} R_{tr}(\mathbf{w})$, if $||\mathbf{m} \odot \mathbf{w}^*||_2 \neq 0$, there exist OOD nodes with unbounded error, i.e., $\exists v$ s.t. $\lim_{||\mathbf{d}_{v,2}|| \to \infty} (\hat{y}_v - y_v)^2 = \infty$.*

The proposition 1 shows that a classifier trained with masks and empirical risk minimization has unbounded risks in testing data under distribution shifts as the classifier uses variant patterns to make predictions. Next, we show that under mild conditions, an invariant linear classifier in the spectral

domain can solve this problem. Denote $\mathbf{\Phi} \in \mathbb{C}^{T \times T}$ as the Fourier bases, where $\mathbf{\Phi}_{k,t} = \frac{1}{\sqrt{T}}e^{-j\frac{2\pi kt}{T}}$. Denote $\mathbf{z}_v = \mathbf{\Phi}\sum_{u \in \mathcal{N}_v} 1$ as the spectral representation after a linear message-passing. The prediction is $\hat{y}_v = \mathbf{w}^{\mathrm{H}}(\mathbf{m} \odot \mathbf{z}_v)$, where $\mathbf{m} \in \mathbb{C}^{T \times 1}$ is the mask to filter the spectral patterns, $\mathbf{w} \in \mathbb{C}^{T \times 1}$ is a linear classifier, and $(\cdot)^{\mathrm{H}}$ denotes Hermitian transpose. We have the following proposition.

**Proposition 2** *If* $\left(\overline{\mathbf{\Phi}\mathbf{d}_{v,1}} \odot \mathbf{\Phi}\mathbf{d}_{v,1}\right) \odot \left(\overline{\mathbf{\Phi}\mathbf{d}_{v,2}} \odot \mathbf{\Phi}\mathbf{d}_{v,2}\right) = \mathbf{0}, \forall \mathbf{d}_{v,1}, \mathbf{d}_{v,2},$ *then* $\exists \mathbf{m} \in \mathbb{C}^{T \times 1}$ *such that the optimal spectral classifier in the training data has bounded error, i.e., for* $\mathbf{w}^* = \arg\min_{\mathbf{w}} R_{tr}(\mathbf{w}), \exists \epsilon > 0, \forall v, \lim_{||\mathbf{d}_{\mathbf{v,2}}|| \to \infty}(\hat{y}_v - y_v)^2 < \epsilon.$

The proposition 2 shows that if the frequency bandwidths of invariant and variant patterns do not have any overlap, there exists a spectral mask such that a linear classifier trained with empirical risk minimization in the spectral domain will have bounded risk in any testing data distribution. This example motivates us to capture invariant and variant patterns in the spectral domain, which is not feasible in the spatial-temporal domain.

# 4  Method

In this section, we introduce our method named Spectral Invariant Learning for Dynamic Graphs under Distribution Shifts (**SILD**) to handle distribution shifts in dynamic graphs, including three modules, dynamic graph neural networks with Fourier transform, disentangled spectrum mask, and invariant spectral filtering. The framework of our method is shown in Figure 2.

## 4.1  Dynamic Graph Neural Network with Spectral Transform

**Dynamic Graph Trajectories Modeling**  Each node on the dynamic graph has its ego-graph trajectory evolving through time that may determine the node properties or the occurrence of future links. Following [23, 24, 25], we adopt a message-passing network for each graph snapshot to aggregate the neighborhood information at the current time, *i.e.*,

$$\mathbf{m}_{u \to v}^t \leftarrow \mathrm{MSG}(\mathbf{h}_u^t, \mathbf{h}_v^t), \mathbf{h}_v^t \leftarrow \mathrm{AGG}(\{\mathbf{m}_{u \to v}^t \mid u \in \mathcal{N}^t(v)\}, \mathbf{h}_v^t), \tag{1}$$

where 'MSG' and 'AGG' denote message and aggregation functions, $\mathbf{m}_{u \to v}^t$ is the message from node $u$ to node $v$, $\mathbf{h}_u^t$ is the node embedding for node $u$ at time $t$, $\mathcal{N}^t(v) = \{u \mid (u,v) \in \mathcal{E}^t\}$ is node $v$'s neighborhood at time $t$. To model the high-order neighborhood information, we can stack multiple message-passing layers. In this way, the node embedding along time $\{\mathbf{h}_u^t\}_{t=1}^T$ summarizes the evolution of node $u$'s ego-graph trajectories. We denote $\mathbf{H} \in \mathbb{R}^{T \times N \times d}$ as the ego-graph trajectory signals for all nodes on the dynamic graph, where $T$ denotes the total time length, $N$ denotes the number of nodes and $d$ denotes the hidden dimensionality.

**Spectral Transform**  As some patterns on dynamic graphs are unobservable in the time domain, while observable in the spectral domain, we transform the summarized ego-graph trajectory signals $\mathbf{H}$ into the spectral domain via Fourier transform for each node and hidden dimension, *i.e.*,

$$\mathbf{\Phi}_{k,t} = \frac{1}{\sqrt{T}}e^{-j\frac{2\pi kt}{T}}, \mathbf{Z} = \mathbf{\Phi}\mathbf{H}, \tag{2}$$

where $\mathbf{\Phi} \in \mathbb{C}^{K \times T}$ denotes the Fourier bases, $K$ denotes the number of frequency components, and $\mathbf{Z} \in \mathbb{C}^{K \times N \times d}$ denote the node embeddings along frequency components in the spectral domain, and $\mathbf{Z}_{k,n,m} = \sum_{t=1}^T \mathbf{\Phi}_{k,t}\mathbf{H}_{t,n,m}$. By choosing the Fourier bases, our spectral transform has the following advantages: 1) we can use fast Fourier transform (FFT) [26] to accelerate the computation. The computation complexity of Eq. (2) can be reduced from $O(NdT^2)$ to $O(NdTlogT)$. 2) Each basis has clear semantics, *e.g.*, $\mathbf{Z}_k$ denotes the node embeddings at the $k$-th frequency component in the spectral domain. In this way, we can observe how the nodes on the dynamic graph evolve in different frequency bandwidths. 3) Fourier transform is able to capture global and periodic patterns [27, 28], which are common in real-world dynamic graphs, *e.g.*, the interactions on e-commerce networks may result from seasonal sales or product service cycles.

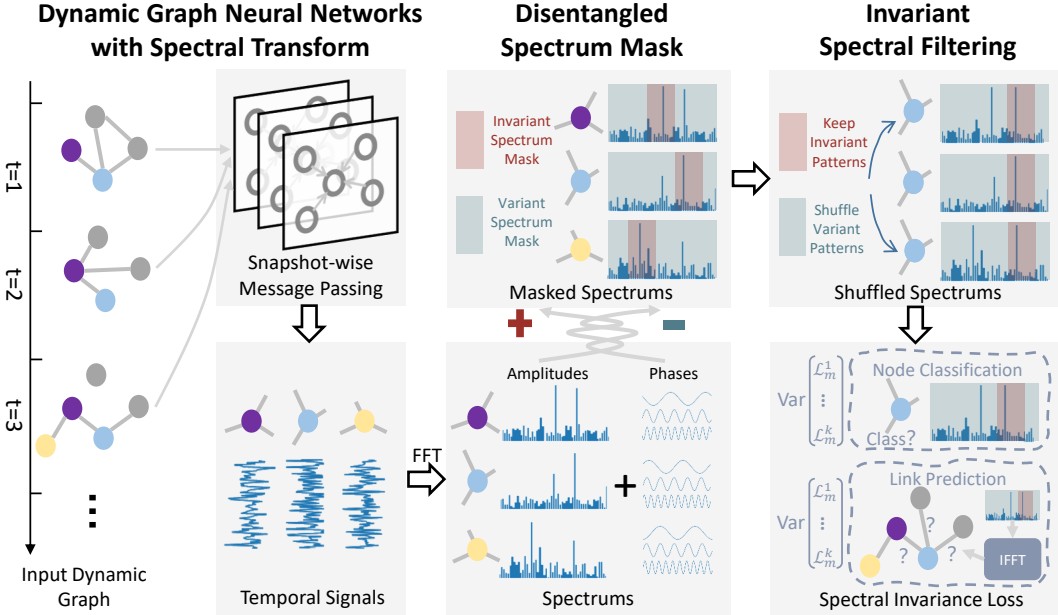

Figure 2: The framework of our proposed method **SILD**. Given a dynamic graph evolving through time, the dynamic graph neural networks with spectral transform first obtain the ego-graph trajectory spectrums in the spectral domain. Then the disentangled spectrum mask leverages the amplitude and phase information of the ego-graph trajectory spectrums to obtain invariant and variant spectrum masks. Last, invariant spectral filtering discovers the invariant and variant patterns via the disentangled spectrum masks, and minimizes the variance of predictions with exposure to various variant patterns, to help the model exploit invariant patterns to make predictions under distribution shifts.

## 4.2 Disentangled Spectrum Mask

To capture invariant patterns in the spectral domain, we propose to explicitly learn spectrum masks to disentangle the invariant and variant patterns. The embeddings in the spectral domain contain the amplitude information as well as the phase information for each node

$$\text{Amp}(\mathbf{Z}) = \left(\text{Imag}^2(\mathbf{Z}) + \text{Real}^2(\mathbf{Z})\right)^{\frac{1}{2}}, \phi(\mathbf{Z}) = \arctan\frac{\text{Imag}(\mathbf{Z})}{\text{Real}(\mathbf{Z})}, \tag{3}$$

where $\text{Real}(\cdot)$ and $\text{Imag}(\cdot)$ denote the real and imaginary part of the complex number, *i.e.*, $\mathbf{Z} = \text{Real}(\mathbf{Z}) + j\text{Imag}(\mathbf{Z})$, $j$ denotes the imaginary unit, $\text{Amp}(\mathbf{Z}) \in \mathbb{R}^{K \times N \times d}$ and $\phi(\mathbf{Z}) \in \mathbb{R}^{K \times N \times d}$ denote the amplitude and phase information respectively. For brevity, the tensor operators in Eq. (3) are all element-wise, *e.g.*, $(\text{Imag}^2(\mathbf{Z}))_{i,j,k} = (\text{Imag}(\mathbf{Z}_{i,j,k}))^2$. Then, we obtain the spectrum masks by leveraging both the amplitude and phase information

$$\mathbf{M} = \text{MLP}(\text{Real}(\mathbf{Z})||\text{Imag}(\mathbf{Z})), \mathbf{M}_I = \text{sigmoid}(\mathbf{M}/\tau), \mathbf{M}_V = \text{sigmoid}(-\mathbf{M}/\tau), \tag{4}$$

where MLP denotes multi-layer perceptrons, $\tau$ is the temperature, $\mathbf{M}_I \in [0,1]^K$ and $\mathbf{M}_V \in [0,1]^K$ denote the spectrum mask for invariant and variant patterns, and $||$ represents the concatenation of the embeddings. In this way, the invariant and variant masks have a negative relationship, and each node can have its own spectrum mask. As the phase information includes high-level semantics in the original signals [29, 30, 31, 32, 33], we keep the phase information unchanged to reduce harm in the fine-grained semantic information for the graph trajectories, and filter the spectrums by the learned disentangled masks in terms of amplitudes,

$$\mathbf{Z}_I = \left(\mathbf{M}_I \odot \text{Amp}(\mathbf{Z})\right) \odot (\cos\phi(\mathbf{Z}) + j\sin\phi(\mathbf{Z})), \mathbf{Z}_V = \left(\mathbf{M}_V \odot \text{Amp}(\mathbf{Z})\right)(\cos\phi(\mathbf{Z}) + j\sin\phi(\mathbf{Z})), \tag{5}$$

where $\mathbf{Z}_I$ and $\mathbf{Z}_V$ denote the summarized invariant and variant patterns in the spectral domain. For node classification tasks, we can directly adopt the spectrums for the classifier to predict classes. For link prediction tasks, we can utilize inverse fast Fourier transform (IFFT) to transform the embeddings into the temporal domain for future link prediction

$$\mathbf{H}'_I = \mathbf{\Phi}^{\text{H}}\mathbf{Z}_I, \mathbf{H}'_V = \mathbf{\Phi}^{\text{H}}\mathbf{Z}_V, \tag{6}$$

where $(\cdot)^{\mathrm{H}}$ is Hermitian transpose, $\mathbf{H}'_I$ and $\mathbf{H}'_V$ denote the filtered invariant and variant patterns that are transformed back into the temporal domain respectively.

### 4.3 Invariant Spectral Filtering

Under distribution shifts, the variant patterns on dynamic graphs have varying relationships with labels, while the invariant patterns have sufficient predictive abilities with regard to labels. We propose invariant spectral filtering to capture the invariant and variant patterns in the spectral domain, and help the model focus on invariant patterns to make predictions, thus handling distribution shifts. We take node classification tasks for an example as follows.

Let $\mathbf{Z}_I \in \mathbb{C}^{K \times N \times d}$ and $\mathbf{Z}_V \in \mathbb{C}^{K \times N \times d}$ be the filtered invariant and variant spectrums in the spectral domain. Then we can utilize the invariant and variant node spectrums to calculate the task loss

$$\mathcal{L}_I = l(f_I(\mathbf{Z}_I), \mathbf{Y}), \mathcal{L}_V = l(f_V(\mathbf{Z}_V), \mathbf{Y}), \tag{7}$$

where $f_I(\cdot)$ and $f_V(\cdot)$ are the classifiers for invariant and variant patterns respectively, $\mathbf{Y}$ is the labels, and $l$ is the loss function. The task loss is utilized to capture the patterns with the predictive abilities of labels. Recall in Assumption 1, the influence of variant patterns on labels is shielded given invariant patterns as the invariant patterns have sufficient predictive abilities w.r.t labels, and thus the model's predictions should not change when being exposed to different variant patterns and the original invariant patterns. Inspired by [15, 7], we calculate the invariance loss by

$$\mathcal{L}_{INV} = \mathrm{Var}(\{\mathcal{L}_m \mid \tilde{\mathbf{z}} : \tilde{\mathbf{z}} \in \mathcal{S}\}), \tag{8}$$

where $\mathcal{L}_m \mid \tilde{\mathbf{z}}$ denotes the mixed loss to measure the model's prediction ability with exposure to the specific variant pattern $\tilde{\mathbf{z}} \in \mathbb{C}^{K \times d}$ that is sampled from a set of variant patterns $\mathcal{S}$. We adopt all the node embeddings in $\mathbf{Z}_V$ to construct the set of variant patterns $\mathcal{S}$. Inspired by [34, 15], we calculate the mixed loss as

$$\mathcal{L}_m \mid \tilde{\mathbf{z}} = l(f_I(\mathbf{Z}_I) \odot \sigma(f_V(\tilde{\mathbf{z}})), \mathbf{Y}), \tag{9}$$

where $\sigma$ denotes the sigmoid function. Then, the final training objective is

$$\min_{\theta, f_I} \mathcal{L}_I + \lambda \mathcal{L}_{INV} + \min_{f_V} \mathcal{L}_V, \tag{10}$$

where $\theta$ is the parameters that encompass all the model parameters except the classifiers, $\lambda$ is a hyperparameter to balance the trade-off between the model's predictive ability and invariance properties. A larger $\lambda$ encourages the model to capture patterns with better invariance under distribution shifts, with the potential risk of lower predictive ability during training, as the shortcuts brought by the variant patterns might be discarded in the training process. After training, we only adopt invariant patterns to make predictions in the inference stage. The overall algorithm for training on node classification datasets is summarized in Algo. 1.

---

**Algorithm 1** Training pipeline for **SILD** on node classification datasets

---

**Require:** Training epochs $L$, sample number $S$, hyperparameter $\lambda$
 1: **for** $l = 1, \ldots, L$ **do**
 2:     Obtain the node embeddings $\mathbf{H}$ with snapshot-wise message passing as Eq. (1)
 3:     Transform the node embeddings into the spectral domain with FFT as Eq. (2)
 4:     Calculate the disentangled spectrum masks as Eq. (4)
 5:     Filter spectrums into invariant and variant patterns as Eq. (5)
 6:     Calculate the task loss as Eq. (7)
 7:     Sample $S$ variant patterns from collections of $\mathbf{Z}_V$ and calculate the invariance loss as Eq. (8)
 8:     Update the model according to Eq. (10)
 9: **end for**

---

## 5 Experiments

In this section, we conduct extensive experiments to verify that our proposed method can handle distribution shifts on dynamic graphs by discovering and utilizing invariant patterns in the spectral domain. More details of the settings and other results can be found in Appendix.

**Baselines.** We adopt several representative dynamic GNNs and Out-of-Distribution(OOD) generalization methods as our baselines:

Table 1: Results of different methods on real-world link prediction and node classification datasets. The best results are in bold and the second-best results are underlined. The year in the Aminer dataset denotes the test split, *e.g.*, 'Aminer15' denotes the average test accuracy in 2015.

| Task | Link Prediction (AUC%) | | Node Classification (ACC%) | | |
|---|---|---|---|---|---|
| Dataset | Collab | Yelp | Aminer15 | Aminer16 | Aminer17 |
| GCRN | $69.72_{\pm0.45}$ | $54.68_{\pm7.59}$ | $47.96_{\pm1.12}$ | $51.33_{\pm0.62}$ | $42.93_{\pm0.71}$ |
| EGCN | $76.15_{\pm0.91}$ | $53.82_{\pm2.06}$ | $44.14_{\pm1.12}$ | $46.28_{\pm1.84}$ | $37.71_{\pm1.84}$ |
| DySAT | $76.59_{\pm0.20}$ | $66.09_{\pm1.42}$ | $48.41_{\pm0.81}$ | $49.76_{\pm0.96}$ | $42.39_{\pm0.62}$ |
| IRM | $75.42_{\pm0.87}$ | $56.02_{\pm16.08}$ | $48.44_{\pm0.13}$ | $50.18_{\pm0.73}$ | $42.40_{\pm0.27}$ |
| VREx | $76.24_{\pm0.77}$ | $66.41_{\pm1.87}$ | $48.70_{\pm0.73}$ | $49.24_{\pm0.27}$ | $42.59_{\pm0.37}$ |
| GroupDRO | $76.33_{\pm0.29}$ | $66.97_{\pm0.61}$ | $48.73_{\pm0.61}$ | $49.74_{\pm0.26}$ | $42.80_{\pm0.36}$ |
| DIDA | $\underline{81.87}_{\pm0.40}$ | $\underline{75.92}_{\pm0.90}$ | $\underline{50.34}_{\pm0.81}$ | $\underline{51.43}_{\pm0.27}$ | $\underline{44.69}_{\pm0.06}$ |
| **SILD** | $\mathbf{84.09}_{\pm\mathbf{0.16}}$ | $\mathbf{78.65}_{\pm\mathbf{2.22}}$ | $\mathbf{52.35}_{\pm\mathbf{1.04}}$ | $\mathbf{54.11}_{\pm\mathbf{0.62}}$ | $\mathbf{45.54}_{\pm\mathbf{1.19}}$ |

Table 2: Results of different methods on synthetic link prediction and node classification datasets. The best results are in bold and the second-best results are underlined. A larger 'shift' denotes a higher distribution shift level.

| Dataset | Link-Synthetic (AUC%) | | | Node-Synthetic (ACC%) | | |
|---|---|---|---|---|---|---|
| Shift | 0.4 | 0.6 | 0.8 | 0.4 | 0.6 | 0.8 |
| GCRN | $72.57_{\pm0.72}$ | $72.29_{\pm0.47}$ | $67.26_{\pm0.22}$ | $27.19_{\pm2.18}$ | $25.95_{\pm0.80}$ | $29.26_{\pm0.69}$ |
| EGCN | $69.00_{\pm0.53}$ | $62.70_{\pm1.14}$ | $60.13_{\pm0.89}$ | $24.01_{\pm2.29}$ | $22.75_{\pm0.96}$ | $24.98_{\pm1.32}$ |
| DySAT | $70.24_{\pm1.26}$ | $64.01_{\pm0.19}$ | $62.19_{\pm0.39}$ | $40.95_{\pm2.89}$ | $37.94_{\pm1.01}$ | $30.90_{\pm1.97}$ |
| IRM | $69.40_{\pm0.09}$ | $63.97_{\pm0.37}$ | $62.66_{\pm0.33}$ | $33.23_{\pm4.70}$ | $30.29_{\pm1.71}$ | $29.43_{\pm1.38}$ |
| VREx | $70.44_{\pm1.08}$ | $63.99_{\pm0.21}$ | $62.21_{\pm0.40}$ | $41.78_{\pm1.30}$ | $38.11_{\pm2.81}$ | $29.56_{\pm0.44}$ |
| GroupDRO | $70.30_{\pm1.23}$ | $64.05_{\pm0.21}$ | $62.13_{\pm0.35}$ | $41.35_{\pm2.19}$ | $35.74_{\pm3.93}$ | $\underline{31.03}_{\pm1.24}$ |
| DIDA | $\underline{85.20}_{\pm0.84}$ | $\underline{82.89}_{\pm0.23}$ | $\underline{72.59}_{\pm3.31}$ | $\underline{43.33}_{\pm7.74}$ | $\underline{39.48}_{\pm7.93}$ | $28.14_{\pm3.07}$ |
| **SILD** | $\mathbf{85.95}_{\pm\mathbf{0.18}}$ | $\mathbf{84.69}_{\pm\mathbf{1.18}}$ | $\mathbf{78.01}_{\pm\mathbf{0.71}}$ | $\mathbf{43.62}_{\pm\mathbf{2.74}}$ | $\mathbf{39.78}_{\pm\mathbf{3.56}}$ | $\mathbf{38.64}_{\pm\mathbf{2.76}}$ |

- Dynamic GNNs: **GCRN** [35] is a representative dynamic GNN that first adopts a GCN[36] to obtain node embeddings and then a GRU [37] to model the network evolution. **EGCN** [25] adopts an LSTM [38] or GRU [37] to flexibly evolve the GCN [36] parameters through time. **DySAT** [24] aggregates neighborhood information at each graph snapshot using structural attention and models network dynamics with temporal self-attention.

- OOD generalization methods: **IRM** [20] aims at learning an invariant predictor which minimizes the empirical risks for all training domains. **GroupDRO** [39] puts more weight on training domains with larger errors to minimize the worst-group risks across training domains. **V-REx** [40] reduces the differences in the risks across training domains to reduce the model's sensitivity to distributional shifts. As these methods are not specifically designed for dynamic graphs, we adopt the best-performed dynamic GNNs as their backbones on each dataset.

- OOD generalization methods for dynamic graphs: **DIDA** [7] utilizes disentangled attention to capture invariant and variant patterns in the spatial-temporal domain, and conducts spatial-temporal intervention mechanism to let the model focus on invariant patterns to make predictions.

## 5.1 Real-world Datasets

**Settings** We use 3 real-world dynamic graph datasets, including Collab [41, 7], Yelp [24, 7] and Aminer [42, 43]. Following [7], we adopt the challenging inductive future link prediction task on Collab and Yelp, where the model should exploit historical graphs to predict the occurrence of links in the next time step. To measure the model's performance under distribution shifts, the model is tested on another dynamic graph with different fields, which is unseen during training. For node classification, we adopt Aminer, a citation network, where nodes represent papers, and edges from $u$ to $v$ with timestamp $t$ denote the paper $u$ published at year $t$ cites the paper $v$. The task is to predict the venues of the papers. We train on papers published between 2001 - 2011, validate on

those published in 2012 - 2014, and test on those published since 2015. On this dataset, the model is tested to exploit the invariant patterns and make stable predictions under distribution shifts, where the patterns on the dynamic graph may vary in different years.

**Results** Based on the results in Table 1, we have the following observations: 1) *Under distribution shifts, the general OOD generalization baselines have limited improvements over the dynamic GNNs*, *e.g.*, GroupDRO improves over DySAT with 0.9% in Yelp and 0.3% in Aminer15 respectively. A plausible reason is that they are not specially designed to handle distribution shifts on dynamic graphs, and may not consider the graph structural and temporal dynamics to capture invariant patterns. Another reason might be that they strongly rely on high-quality environment labels to capture invariant patterns, which are almost unavailable on real-world dynamic graphs. 2) *Our method can better handle distribution shifts than the baselines.* The datasets have strong distribution shifts, *e.g.*, COVID-19 happens midway and has considerable influence on the consumer behavior on Yelp, and the citation patterns may shift with the outbreak of deep neural networks on Aminer. Nevertheless, our method **SILD** has significant improvements over the state-of-the-art OOD generalization baseline for dynamic graphs DIDA on all datasets, *e.g.*, 2% on average for most datasets, which verifies that our method can better capture the invariant and variant patterns in the spectral domain, and thus handling distribution shifts on dynamic graphs.

## 5.2 Synthetic Datasets

**Settings** To evaluate the model's generalization ability under distribution shifts, we conduct experiments on synthetic link prediction and node classification datasets, which are constructed by introducing manually-designed distribution shifts. For link prediction datasets, we follow [7] to generate additional varying features for each node and timestamps on the original dataset Collab, where these additional features are constructed with spurious correlations w.r.t the labels, *i.e.*, the links in the next timestamps. The spurious correlation degree is determined by a shift level parameter. On this dataset, to have better generalization ability, the model should not rely on variant patterns that exploit the additional features with spurious correlations. For node classification, we briefly introduce the construction of the synthetic dataset as follows. We generate the dynamic graph with stochastic block model [44], where the link probability between nodes at each graph snapshot is determined by two frequency factors. The correlation of one of the factors with class labels is always 1, while the other factor has a variant relationship with labels, where the relationship is also controlled by a shift level parameter. The model should discover and focus on the invariant frequency factors whose relationship with labels is invariant under distribution shifts. For both datasets, we set the shift level parameters as 0.4, 0.6, 0.8 for training and validation splits, and 0 for test splits.

**Results** Based on the results in Table 2, we have the following observations: 1) *Our method can better handle distribution shifts than the baselines, especially under stronger distribution shifts.* **SILD** consistently outperforms DyGNN and general OOD generalization baselines by a significantly large margin, which can credit to our special design to handle distribution shifts on dynamic graphs in the spectral domain. Our method also has a significant improvement over the best-performed baseline under the strongest distribution shift, *e.g.*, with absolute improvements of 5% in Link-Synthetic(0.8) and 7% in Node-Synthetic(0.8) respectively. 2) *Our method can exploit invariant patterns to consistently alleviate the harmful effects of variant patterns under different distribution shift levels.* As the distribution shift level increases, almost all methods decline in performance since the relationship between variant patterns and labels goes stronger, so that the variant patterns are much easier to be exploited by the model, misleading the training process. However, the performance drop of **SILD** is significantly lower than baselines, which demonstrates that our method can alleviate the harmful effects of variant patterns under distribution shifts by exploiting invariant patterns in the spectral domain.

## 5.3 Ablation Studies

We conduct ablation studies to verify the effectiveness of the proposed disentangled spectrum mask and invariant spectral filtering in **SILD**. The ablated version 'SILD w/o I' removes invariant spectral filtering in **SILD** by setting $\lambda = 0$, and 'SILD w/o M' is trained without the disentangled spectrum masks. From Figure 3, we have the following observations. First, our proposed **SILD** outperforms all the variants as well as the best-performed baseline on all datasets, demonstrating the effectiveness of

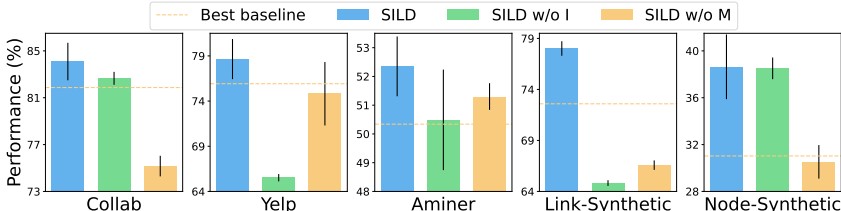

Figure 3: Results of ablation studies, where 'w/o I' removes invariant spectral filtering in **SILD**, 'w/o M' removes disentangled spectrum masks, and 'Best baseline' denotes the best-performed baseline on each dataset. The error bars report the standard deviations. (Best viewed in color)

each component of our proposed method. Second, 'SILD w/o I' and 'SILD w/o M' drop drastically in performance on all datasets compared to the full version, which verifies that our proposed disentangled spectrum mask and spectral invariant learning can help the model to focus on invariant patterns to make predictions and significantly improve the performance under distribution shifts.

# 6  Related Works

**Dynamic Graph Neural Networks**   Dynamic graphs ubiquitously exist in real-world applications [45, 46, 47, 48, 49, 50, 51, 52, 53] such as event forecasting, recommendation, etc. In comparison with static graphs [54, 55, 56, 57, 58, 59], dynamic graphs contain rich temporal information. Considerable research attention has been devoted to dynamic graph neural networks (DyGNNs) [1, 2, 60] to model the complex graph dynamics that include structures and features evolving through time. Some works adopt GNN to aggregate neighborhood information for each graph snapshot, and then utilize a sequence module to model the temporal information [61, 62, 63, 35, 24]. Some others utilize time-encoding techniques to encode the temporal links into time-aware embeddings and adopt a GNN or memory module [64, 65, 66, 67] to process structural information. Some other related works leverage spectral graph neural networks [68], global graph framelet convolution [69], and graph wavelets [70] to obtain better dynamic graph representations. However, distribution shifts remain largely unexplored in dynamic graph neural networks literature. The sole prior work DIDA [7] handles spatial-temporal distribution shifts on dynamic graphs in the temporal domain. To the best of our knowledge, this is the first study of handling distribution shifts on dynamic graphs in the spectral domain.

**Out-of-Distribution Generalization**   A significant proportion of existing machine learning methodologies operate on the assumption that training and testing data are independent and identically distributed (i.i.d.). However, this assumption may not always hold true, especially in the context of complex real-world scenarios [71], and the uncontrollable distribution shifts between training and testing data distribution may lead to a significant decline in the model performance. Out-of-Distribution (OOD) generalization problem has recently drawn great attention in various areas [72, 71, 73]. Some works handle structural distribution shifts on static graphs [74, 15, 16, 75, 5, 76, 77, 78, 79, 80, 81] and temporal distribution shifts on time-series data [11, 12, 6, 13, 14, 82]. However, how to handle distribution shifts on dynamic graphs in the spectral domain remains unexplored.

**Spectral Methods in Neural Networks**   The applications of spectral methods in neural networks have been broadly explored in many areas, including static graph data [83, 84, 85, 86, 87], time-series data [68, 88, 89, 90, 91], etc., for their advantages of modeling global patterns, powerful expressiveness and interpretability [92, 28]. Some work [93] proposes to reconstruct the image in the spectral domain to obtain robust image representations. Some work [29] proposes to augment the image data by perturbing the amplitude information in the spectral domain. Some work [94] proposes a multiwavelet-based method for compressing operator kernels. However, these methods are not applicable to dynamic graphs, not to mention the more complex scenarios under distribution shifts.

# 7 Conclusion

In this paper, we propose a novel model named Spectral Invariant Learning for Dynamic Graphs under Distribution Shifts (**SILD**), which can handle distribution shifts on dynamic graphs in the spectral domain. We design a DyGNN with Fourier transform to obtain the ego-graph trajectory spectrums. Then we propose a disentangled spectrum mask and invariant spectral filtering to discover the invariant and variant patterns in the spectral domain, and help the model rely on invariant spectral patterns to make predictions. Extensive experimental results on several synthetic and real-world datasets, including both node classification and link prediction tasks, demonstrate the superior performance of our method compared to state-of-the-art baselines under distribution shifts. One limitation is that in this paper we mainly focus on dynamic graphs in scenarios of discrete snapshots, and we leave extending our methods to continuous dynamic graphs for further explorations.

## Acknowledgements

This work was supported by the National Key Research and Development Program of China No. 2020AAA0106300, National Natural Science Foundation of China (No. 62222209, 62250008, 62102222, 62206149), Beijing National Research Center for Information Science and Technology under Grant No. BNR2023RC01003, BNR2023TD03006, China National Postdoctoral Program for Innovative Talents No. BX20220185, China Postdoctoral Science Foundation No. 2022M711813, and Beijing Key Lab of Networked Multimedia. All opinions, findings, conclusions, and recommendations in this paper are those of the authors and do not necessarily reflect the views of the funding agencies.

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
