# Spectral Invariant Learning for Dynamic Graphs under Distribution Shifts
## (Appendix)

**Zeyang Zhang**[1]*, **Xin Wang**[1]†, **Ziwei Zhang**[1], **Zhou Qin**[2],
**Weigao Wen**[2], **Hui Xue**[2], **Haoyang Li**[1], **Wenwu Zhu**[1]†

[1]Department of Computer Science and Technology, BNRist, Tsinghua University, [2]Alibaba Group
zy-zhang20@mails.tsinghua.edu.cn, {xin_wang, zwzhang}@tsinghua.edu.cn,
{qinzhou.qinzhou, weigao.wen, hui.xueh}@alibaba-inc.com,
lihy18@mails.tsinghua.edu.cn, wwzhu@tsinghua.edu.cn

## A  Notations

Table 1: The summary of the notations and their descriptions

| Notations | Descriptions |
|---|---|
| $\mathcal{G} = (\mathcal{V}, \mathcal{E})$ | A graph with the node set and edge set |
| $\mathcal{G}^t = (\mathcal{V}^t, \mathcal{E}^t)$ | Graph slice at time $t$ |
| $\mathbf{X}^t, \mathbf{A}^t$ | Features and adjacency matrix of a graph at time $t$ |
| $\mathcal{G}^{1:t}, Y^t, \mathbf{G}^{1:t}, \mathbf{Y}^t$ | Graph trajectory, label and their corresponding random variable |
| $\mathcal{G}_v^{1:t}, y_v^t, \mathbf{G}_v^{1:t}, \mathbf{y}_v^t$ | Ego-graph trajectory, the node's label and their corresponding random variable |
| $p(\cdot)$ | Probability distribution |
| $P, \mathbf{P}$ | Pattern and its corresponding random variable |
| $\mathbf{d}_{v,1}, \mathbf{d}_{v,2}$ | The degrees of node $v$ varying by time |
| $\mathbf{g}, \mathbf{w}$ | The parameters of linear classifiers |
| $\mathbf{m}, \mathbf{M}$ | The mask to filter node representations |
| $\hat{y}_v$ | The prediction for the node $v$ |
| $R_{tr}(\mathbf{w})$ | The risks of the classifier $\mathbf{w}$ in training data |
| $\mathbf{\Phi}$ | The Fourier bases |
| $\overline{x}$ | The conjugate of $x$ |
| $\mathrm{MSG}(\cdot), \mathrm{AGG}(\cdot)$ | Message and Aggregation functions |
| $\mathbf{h}_u^t$ | Hidden embeddings for node $u$ at time $t$ |
| $\mathbf{H}, \mathbf{Z}$ | Node representations in the temporal domain and spectral domain |
| $d$ | The dimensionality of node representations |
| $\mathrm{Amp}(\mathbf{Z}), \phi(\mathbf{Z})$ | The amplitudes and phases of the representations $\mathbf{Z}$ |
| $\mathbf{x}^H$ | The Hermitian transpose of $\mathbf{x}$ |
| $T, K$ | The number of time stamps and the number of frequency components |
| $f(\cdot)$ | Predictors |
| $\ell$ | Loss function |
| $\mathcal{L}, \mathcal{L}_m, \mathcal{L}_{INV}$ | Task loss, mixed loss and invariance loss |

---

*This work was done during the author's internship at Alibaba Group
†Corresponding authors

37th Conference on Neural Information Processing Systems (NeurIPS 2023).

## B  Theoretical Analyses

### B.1  Proof of Proposition 1

**Proposition 1.** *For any mask* $\mathbf{m} \in \mathbb{R}^{T \times 1}$, *for the optimal classifier in the training data* $\mathbf{w}^* = \arg\min_{\mathbf{w}} R_{tr}(\mathbf{w})$, *if* $||\mathbf{m} \odot \mathbf{w}^*||_2 \neq 0$, *there exist OOD nodes with unbounded error, i.e.,* $\exists v$ *s.t.* $\lim_{||\mathbf{d}_{v,2}|| \to \infty} (\hat{y}_v - y_v)^2 = \infty$.

*Proof.* For any mask $\mathbf{m} \in \mathbb{R}^{T \times 1}$, the predictions of the model is

$$\hat{y}_v = \mathbf{w}^{\top}(\mathbf{m} \odot (\mathbf{d}_{v,1} + \mathbf{d}_{v,2})). \tag{1}$$

We assume that $||\mathbf{m} \odot \mathbf{w}|| \neq 0$, otherwise the classifier is a trivial solution and always predicts $\hat{y}_v = 0$ for any node $v$. The empirical risk in training data is

$$R_{tr}(\mathbf{w}) = \frac{1}{|D_{tr}|} \sum_{v \in D_{tr}} (\hat{y}_v - y_v)^2. \tag{2}$$

By setting $\frac{\partial R_{tr}(\mathbf{w})}{\partial \mathbf{w}} = 0$, we have the optimal classifier learned from the training data

$$\mathbf{w}^* = \frac{\mathbf{m} \odot \sum_{v \in D_{tr}} (\mathbf{d}_{v,1} + \mathbf{d}_{v,2}) \mathbf{g}^{\top} \mathbf{d}_{v,1}}{\sum_{v \in D_{tr}} (\mathbf{m} \odot (\mathbf{d}_{v,1} + \mathbf{d}_{v,2}))\top(\mathbf{m} \odot (\mathbf{d}_{v,1} + \mathbf{d}_{v,2}))}. \tag{3}$$

Then for a node $v$ that has variant patterns $\mathbf{d}_{v,2} = \alpha \mathbf{m} \odot \mathbf{w}^*$ and $\alpha \in \mathbb{R}$, the loss of the model's prediction is

$$
\begin{aligned}
l_v = (\hat{y}_v - y_v)^2 &= \left( \mathbf{w}^{*\top}(\mathbf{m} \odot (\mathbf{d}_{v,1} + \mathbf{d}_{v,2})) - \mathbf{g}^{\top} \mathbf{d}_{v,1} \right)^2 \\
&= \left( \alpha ||\mathbf{m} \odot \mathbf{w}^*||^2 + (\mathbf{w}^{*\top} \mathbf{m} \odot \mathbf{d}_{v,1} - \mathbf{g}^{\top} \mathbf{d}_{v,1}) \right)^2 \\
&= \left( ||\mathbf{d}_{v,2}|| ||\mathbf{m} \odot \mathbf{w}^*|| + (\mathbf{w}^{*\top} \mathbf{m} \odot \mathbf{d}_{v,1} - \mathbf{g}^{\top} \mathbf{d}_{v,1}) \right)^2.
\end{aligned}
\tag{4}
$$

Then $\forall \epsilon > 0$, when $||\mathbf{d}_{v,2}|| > \frac{\sqrt{\epsilon} - (\mathbf{w}^{*\top} \mathbf{m} \odot \mathbf{d}_{v,1} - \mathbf{g}^{\top} \mathbf{d}_{v,1})}{||\mathbf{m} \odot \mathbf{w}^*||}$, i.e., $\alpha > \frac{\sqrt{\epsilon} - (\mathbf{w}^{*\top} \mathbf{m} \odot \mathbf{d}_{v,1} - \mathbf{g}^{\top} \mathbf{d}_{v,1})}{||\mathbf{m} \odot \mathbf{w}^*||^2}$, $(\hat{y}_v - y_v)^2 > \epsilon$, indicating that $\lim_{||\mathbf{d}_{v,2}|| \to \infty} (\hat{y}_v - y_v)^2 = \infty$. Thus we conclude the proof. $\square$

### B.2  Proof of Proposition 2

**Proposition 2.** *If* $\left( \overline{\mathbf{\Phi} \mathbf{d}_{v,1}} \odot \mathbf{\Phi} \mathbf{d}_{v,1} \right) \odot \left( \overline{\mathbf{\Phi} \mathbf{d}_{v,2}} \odot \mathbf{\Phi} \mathbf{d}_{v,2} \right) = \mathbf{0}, \forall \mathbf{d}_{v,1}, \mathbf{d}_{v,2}$, *then* $\exists \mathbf{m} \in \mathbb{C}^{T \times 1}$ *such that the optimal spectral classifier in the training data has bounded error, i.e., for* $\mathbf{w}^* = \arg\min_{\mathbf{w}} R_{tr}(\mathbf{w})$, $\exists \epsilon > 0, \forall v, \lim_{||\mathbf{d}_{\mathbf{v},\mathbf{2}}|| \to \infty} (\hat{y}_v - y_v)^2 < \epsilon$.

*Proof.* Let the mask in the spectral domain $\mathbf{m} \in \mathbb{C}^{K \times 1}$ be

$$\mathbf{m}_i = \begin{cases} 0 & \text{if } \exists v, \left( \overline{\mathbf{\Phi} \mathbf{d}_{v,2}} \odot \mathbf{\Phi} \mathbf{d}_{v,2} \right)_i \neq 0 \\ 1 & \text{otherwise} \end{cases}. \tag{5}$$

Since the frequency bandwidths of invariant and variants patterns do not have overlap, i.e., $\left( \overline{\mathbf{\Phi} \mathbf{d}_{v,1}} \odot \mathbf{\Phi} \mathbf{d}_{v,1} \right) \odot \left( \overline{\mathbf{\Phi} \mathbf{d}_{v,2}} \odot \mathbf{\Phi} \mathbf{d}_{v,2} \right) = \mathbf{0}, \forall \mathbf{d}_{v,1}, \mathbf{d}_{v,2}$, we have $\mathbf{m}_i \odot \mathbf{z}_{v,1} = \mathbf{z}_{v,1}$ and $\mathbf{m}_i \odot \mathbf{z}_{v,2}$ for any node $v$. Let $\mathbf{w}_1 = \mathbf{\Phi} \mathbf{g}$, then the prediction for any node $v$ is

$$
\begin{aligned}
\hat{y}_v &= (\mathbf{\Phi} \mathbf{g})^{\mathrm{H}}(\mathbf{m} \odot (\mathbf{z}_{v,1} + \mathbf{z}_{v,2})) \\
&= \mathbf{g}^{\top} \mathbf{\Phi}^{\mathrm{H}}(\mathbf{z}_{v,1}) \\
&= \mathbf{g}^{\top} \mathbf{\Phi}^{\mathrm{H}}(\mathbf{\Phi} \mathbf{d}_{v,1}) \\
&= \mathbf{g}^{\top} \mathbf{d}_{v,1}.
\end{aligned}
\tag{6}
$$

For any node $v$, we have $(\hat{y}_v - y_v)^2 = 0$, so that $\mathbf{w}_1 = \arg\min_{\mathbf{w}} R_{tr}(\mathbf{w})$, and $\forall v, \lim_{||\mathbf{d}_{\mathbf{v},\mathbf{2}}|| \to \infty} (\hat{y}_v - y_v)^2 < 1$. Thus we conclude the proof. $\square$

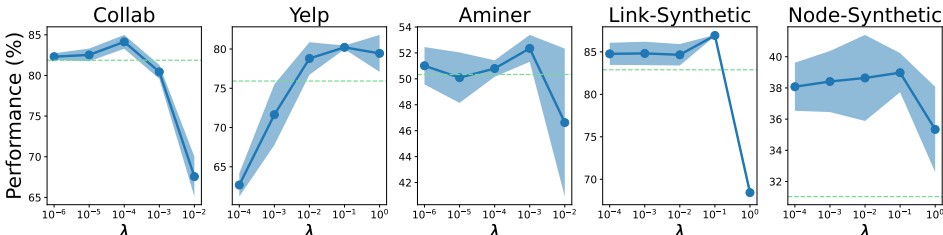

Figure 1: Sensitivity of hyperparameter $\lambda$. The area shows the average AUC and standard deviations in the test stage. The dashed line represents the average AUC of the best performed baseline.

## C  Additional Experiments and Analyses

### C.1  Hyperparameter Sensitivity

We analyze the sensitivity of hyperparameter $\lambda$ in **SILD** for each dataset by altering the hyperparameter on a base ten logarithmic scale. As shown in Figure 1, when the hyperparameter $\lambda$ is too small or too large, the performance of the model deteriorates in most datasets, which verifies that the hyperparameter $\lambda$ is the tradeoff between the sufficiency and invariance conditions of the patterns captured by the model.

### C.2  Complexity Analysis

We analyze the computational complexity of **SILD** as follows. Denote the total number of nodes and edges in the graph as $|\mathcal{V}|$ and $|\mathcal{E}|$, and the dimensionality of the hidden representation as $d$. The snapshot-wise message passing has a time complexity of $O(|\mathcal{E}|d + |\mathcal{V}|d^2)$. The fast Fourier transform has a time complexity of $O(|\mathcal{V}|d \log T)$. The disentangled spectrum mask has a time complexity of $O(|\mathcal{V}|d)$. Denote $|N_p|$ as the number of nodes or edges to predict and $S$ as the sampling number of variant patterns. Our invariant spectral filtering has a time complexity of $O(|N_p|Sd)$ in training, and does not put extra time complexity in inference. Therefore, the overall time complexity of **SILD** is $O(|\mathcal{E}|d + |\mathcal{V}|d^2 + |\mathcal{V}|d + |\mathcal{V}|d \log T + |N_p|Sd)$. In summary, the time complexity of **SILD** has a linear time complexity with respect to the number of nodes and edges, which is on par with the existing dynamic GNNs.

## D  Reproducibility Details

### D.1  Training & Evaluation

**Hyperparameters**  Following [1], for all methods, we adopt the hidden dimension as 32 for Aminer and 16 for other datasets. The number of layers is set to 2, and the models are optimized with the Adam optimizer [2] with a learning rate 1e-2 and weight decay 5e-7. The early stopping strategy on the validation splits is adopted, with 100 epochs for Node-Synthetic datasets and 50 epochs for other datasets. For **SILD**, we set the sampling number of variant patterns as 1000 for Collab and Yelp, and 100 for other datasets, and $\lambda$ as 1e-4,1e-3,1e-2,1e-2,1e-2 for Collab, Aminer, Yelp, Link-Synthetic, and Node-Synthetic datasets respectively.

**Evaluation**  For link prediction tasks, we randomly sample negative links from the nodes that actually do not have links in-between, and the number of negative links is the same as the number of positive links. All the negative and positive samples for validation and testing set are kept the same for all methods. We use the inner product of the two node representations to predict links, use cross-entropy as the loss function $\ell$, and use Area under the ROC Curve (AUC) as the evaluation metric. For node classification tasks, we use a two-layer MLP for the node classifier, use cross-entropy as the loss function $\ell$, and use Accuracy (ACC) as the evaluation metric. We randomly run the experiments three times, and report the average results and standard deviations.

**Details of SILD**  For the node classification dataset Aminer, we conduct the missing graph trajectory complementation as follows. In practice, dynamic graphs usually encounter with the issues

of incomplete trajectories, i.e., the nodes have missing historical trajectories for some reasons. For example, on academic citation networks, the papers on dynamic graphs are always different each year and they only have structures (cite other papers) at the published year, which means that they only have a one-year trajectory. In these cases, the modeling of dynamics would be difficult and also inaccurate. To complement the missing historical graph trajectories, we utilize the current structure as the virtual past structure to help model the neighborhood evolution for the node to predict at $t'$, and the message passing is

$$\mathbf{m}_{u \to v}^t \leftarrow \mathrm{MSG}(\mathbf{h}_u^t, \mathbf{h}_v^t), \mathbf{h}_v^t \leftarrow \mathrm{AGG}(\{\mathbf{m}_{u \to v}^t \mid u \in \mathcal{N}^t(v) \bigcup \mathcal{N}^{t'}(v)\}, \mathbf{h}_v^t}). \tag{7}$$

In this way, the node embedding $\mathbf{h}_u^t$ for node $u$ which appears at time $t \leq t'$ denotes the neighborhood information it may aggregate if it appears at time $t$. Note that in Eq. (7), the target is to predict the node labels at time $t'$, where the current neighborhood $\mathcal{N}^{t'}(v)$ is known to all methods and this method does not exploit extra future information. For the message and aggregation functions, we adopt DIDA [1] for Yelp dataset and GAT [3] for other datasets. We adopt two-layer MLPs for both the invariant and variant node classifiers.

### D.2    Dataset Details

We summarize the dataset statistics in Table 2 and describe the dataset details as follows.

**Collab** [4, 1][3] is an academic collaboration dataset with papers that were published during 1990-2006, where the nodes and edges represent author and coauthorship respectively. The author features are obtained by averaging the embeddings of the author-related papers, which are extracted by word2vec [5] from the paper abstracts. The distribution shift comes from different fields, including "Data Mining", "Database", "Medical Informatics", "Theory" and "Visualization". We use 10,1,5 chronological graph slices for training, validation and testing respectively.

**Yelp** [6, 1][4] is a business review dataset, where the nodes and edges represent customers or businesses and review behaviors respectively. We utilize the data from January 2019 to December 2020, and select users and reviews with interactions of more than 10. We use word2vec [5] to extract 32-dimensional features from the reviews and average to obtain the user and business features. The distribution shift comes from the out-break of COVID-19 midway as well as the different business categories including "Pizza", "American (New) Food", "Coffee & Tea ", "Sushi Bars" and "Fast Food". We use 15,1,8 chronological graph slices for training, validation and testing respectively.

**Aminer** [7, 8] is a citation network extracted from DBLP, ACM, MAG, and other sources. We select the top 20 venues, and the task is to predict the venues of the papers. We use word2vec [5] to extract 128-dimensional features from paper abstracts and average to obtain paper features. The distribution shift may come from the out-break of deep learning. We train on papers published between 2001 - 2011, validate on those published in 2012-2014, and test on those published since 2015.

**Link-Synthetic** [1] introduces manual-designed distribution shift on Collab dataset. Denote the original features and structures in Collab as $\mathbf{X}_1^t$ and structures as $\mathbf{A}^t$. We introduce features $\mathbf{X}_2^t$ with a variable correlation with the labels, which are obtained by training the embeddings $\mathbf{X}_2^t \in \mathbb{R}^{N \times d}$ with the reconstruction loss $\ell(\mathbf{X}_2^t \mathbf{X}_2^{t\top}, \tilde{\mathbf{A}}^{t+1})$, where $\tilde{\mathbf{A}}^{t+1}$ refers to the sampled links, and $\ell$ refers to the cross-entropy loss function. In this way, the generated features can have strong correlations with the sampled links. For each time $t$, we uniformly sample $p(t)|\mathcal{E}^{t+1}|$ positive links and $(1 - p(t))|\mathcal{E}^{t+1}|$ negative links in $\mathbf{A}^{t+1}$ and the sampling probability $p(t) = \mathrm{clip}(\overline{p} + \sigma cos(t), 0, 1)$ refers to the intensity of shifts. By controlling the parameter $p$, we can control the correlations of $\mathbf{X}^t$ and labels $\mathbf{A}^{t+1}$ to vary in training and test stage. Since the model observes the $\mathbf{X}^t = [\mathbf{X}_1^t || \mathbf{X}_2^t]$ simultaneously and the variant features are not marked, the model should discover and get rid of the variant features to handle distribution shifts. Similar to Collab dataset, we use 10,1,5 chronological graph slices for training, validation and test respectively.

**Node-Synthetic** introduces manually designed distribution shifts for node classification tasks, by simulating that some frequency components on dynamic graphs have invariant correlations with labels while some others do not. We adopt a stochastic block model (SBM) [9] to generate links

---

[3]https://www.aminer.cn/collaboration.
[4]https://www.yelp.com/dataset

Table 2: The summary of the dataset statistics.

| Dataset | # Snapshots | # Nodes | # Links | Time Granularity | # Features | Evolving Features |
|---|---|---|---|---|---|---|
| Collab | 16 | 23,035 | 151,790 | Year | 32 | No |
| Yelp | 24 | 13,095 | 65,375 | Month | 32 | No |
| Link-Synthetic | 16 | 23,035 | 151,790 | - | 64 | Yes |
| Aminer | 17 | 43,141 | 851,527 | Year | 128 | No |
| Node-Synthetic | 100 | 5,000 | 11,252,385 | - | 4 | No |

between nodes. For brevity, we denote the SBM model as $\text{SBM}(\mathbf{p}_{in}, p_{out})$, where $\mathbf{p}_{in} \in [0,1]^{C \times 1}$ and $p_{out}$ denotes the link probability between the nodes belonging to the same class and the link probability between the nodes from different classes respectively. We adopt $C = 5$ classes. Based on the class label, each node has two types of parameters $f_{low} \in \{0.02, 0.04, 0.08, 0.10, 0.12\}$ and $f_{high} \in \{0.22, 0.24, 0.28, 0.30, 0.32\}$. The correlation of $f_{low}$ with labels is set to 0.4, 0.6, 0.8 respectively for training and validation and 0 for testing, and the correlation of $f_{high}$ with labels is set to 1 for all data splits. The dynamic graph $\mathcal{G}^t$ at time $t$ is constructed by mixing multiple graphs together, including a random graph $\mathcal{G}_r^t$ generated from Gaussian noises, a graph constructed by the invariant parameter $\mathcal{G}_I^t = \text{SBM}(\mathbf{p}_{in}^{high}(t), p_{out})$ and a graph constructed by the variant parameter $\mathcal{G}_I^t = \text{SBM}(\mathbf{p}_{in}^{low}(t), p_{out})$. The relationship between the parameters and the link probability is $\mathbf{p}_{in}^{low}(t, f) = S_1(2 + \cos(2\pi f t))$ and $\mathbf{p}_{in}^{high}(t, f) = S_2(2 + \cos(2\pi f t))$. We set 1e-3, 1e-2, 5e-3 for $p_{out}$, $S_1$ and $S_2$ respectively. We generate 4-dimensional random features for each node. On this dataset, to have better generalization ability, the model should discover and focus on the dynamic graph constructed with the invariant parameter to make predictions.

### D.3 Configurations

All the experiments are conducted with:

- Operating System: Ubuntu 20.04.5 LTS
- CPU: Intel(R) Xeon(R) Gold 6240 CPU @ 2.60GHz
- GPU: NVIDIA GeForce RTX 3090 with 24 GB of memory
- Software: Python 3.9.12, Cuda 11.3, PyTorch [10] 1.12.1, PyTorch Geometric [11] 2.0.4.