# OpenReview forum: "Spectral Invariant Learning for Dynamic Graphs under Distribution Shifts"
_NeurIPS.cc/2023/Conference — NeurIPS 2023 poster_

### Official Review · Reviewer_QjfT · 2023-07-01

**Soundness:** 1 poor
**Presentation:** 2 fair
**Contribution:** 2 fair
**Rating:** 3
**Confidence:** 4

**Summary:**

This paper studies the distribution shifts, i.e., out-of-distribution generalization, on graph data and focuses on distribution shifts on temporal dynamic graphs. To this end, the authors propose a model from the spectral domain and resort to the established techniques of invariant learning. Experiments on synthetic and real-world data show the effectiveness of the proposed model when compared to a number of baselines.

**Strengths:**

1. The problem of distribution shifts on graph data is significant and a research area of growing interest.

2. The proposed model is well motivated and grounded.

**Weaknesses:**

1. The novelty is limited especially given that the model is based on the existing framework

2. The presentation can be improved as some parts of the model section are unclear and confusing

3. The empirical comparison is insufficient

**Questions:**

1. How the mixed loss L_{INV} is calculated and how the sampling for the variant patterns is conducted?

2. What is the rationale of the objective (10)? Can it guarantee the disentanglement for the invariant/variant frequency components and a solution for OOD generalization?

3. The model seems to need two fundamental assumptions: 1) the observed temporal graphs are generated by a mixture of invariant and variant frequency patterns; 2) the invariant and variant components can be strictly disentangled. The authors are encouraged to discuss more on this, especially for how universal these assumptions can be and what if they are violated in real data.

4. How does the model perform compared to other graph OOD competitors that do not involve temporal modeling?

**Limitations:**

The proposed model is heavily based on the invariant learning framework on graph data, and the originality concerning the distribution shifts seems limited. The novel part from the frequency modeling can be an orthogonal technique that is also applicable for other problems beyond the out-of-distribution generalization, and the authors fail to argue the advantage and uniqueness of the frequency modeling w.r.t. the temporal OOD problem. This weakens the originality.

Another concern to me is the soundness of the model. The proposed method requires the assumptions for the data generation of the temporal graph that may not hold in most cases. Furthermore, even when the assumption holds, it is not clear how the model can guarantee disentanglement of the variant/invariant frequency components and a well-posed solution for OOD generalization.

The comparison is insufficient, especially lacking comparison with some common baselines that do not involve the time information on graph data, e.g., the base model GCN, GraphSAGE, etc., as well as the graph OOD models EERM [1] and DIR [2].

[1] Qitian Wu, Hengrui Zhang, Junchi Yan, and David Wipf. Handling distribution shifts on graphs: An invariance perspective, ICLR 2022.

[2] Ying-Xin Wu, Xiang Wang, An Zhang, Xiangnan He, and Tat-Seng Chua. Discovering invariant rationales for graph neural networks, ICLR 2022.

**Post-rebuttal comments**

After discussions, the paper has the following main issues:

1. The novelty and contributions are limited. The proposed model has no essential difference to the NeurIPS22 work DIDA which has explored invariant learning on dynamic graphs.

2. The motivation examples are flawed, and the model is vague. How the model actually learns invariant/variant frequency patterns is unclear. The theoretical results are trivial and the reproducibility is zero.

3. The comparison and evaluation are limited. Some important OOD baselines are not compared or weakened in the experiments. The datasets are also small with a dozen graph snapshots, arousing a question whether the model can indeed learn complex invariant frequency patterns. There is no concrete verification for this critical point.

4. The model section seems like a combination of existing works, but unfortunately, citations are not properly provided. For example, the model in Sec 4 has large overlap with DIR, the theoretical motivation in Sec 3 is similar to EERM, the experiment in Sec 5 is nearly the same as DIDA.

---

> ### Author Rebuttal · Authors · 2023-08-09
>
> We sincerely appreciate the insightful comments provided by the reviewer. We have carefully considered each point raised and would like to respond as follows:
>
> > How the mixed loss $\mathcal{L}_{INV}$ is calculated and how the sampling for the variant patterns is conducted?
>
> Thank you for your comment. We collect the variant patterns of all nodes to construct the variant pattern set $\mathcal{S}$, and uniformly sample the variant patterns from the set without replacement. For each sampled variant pattern, a mixed loss is calculated by Eq. (9) to measure the model's prediction ability with exposure to the specific variant pattern. Finally, we obtain the invariance loss $\mathcal{L}_{INV}$ by calculating the variance of the mixed loss as Eq. (8). We appreciate your question and will provide further clarification in the revised paper.
>
> > What is the rationale of the objective (10)? Can it guarantee the disentanglement for the invariant/variant frequency components and a solution for OOD generalization?
>
> Thank you for your question. In objective Eq. (10), the task losses $\mathcal{L}\_{I}$ and $\mathcal{L}\_{V}$ are utilized to capture the invariant and variant patterns, and the invariance loss $\mathcal{L}_{INV}$ encourages the model to rely on invariant patterns to make predictions. In this way, the model is encouraged to disentangle the invariant and variant patterns and focus on invariant patterns, whose relationships to labels are invariant, to make predictions, and thus handle distribution shifts. In experiments, we empirically find that our method significantly outperforms the baselines in dynamic graph prediction tasks under distribution shifts. We will provide a more detailed explanation of the rationale behind this objective in the revised paper.
>
> > The authors are encouraged to discuss more on the two fundamental assumptions, especially for how universal these assumptions can be and what if they are violated in real data.
>
> Thank you for highlighting this point. We acknowledge that we follow the invariant learning literature to adopt the assumptions of invariant patterns, and they are common on dynamic graphs. For example, on e-commerce networks, some low-frequency interactions may represent the user's long-term interest, while some high-frequency interactions may result from promotions. For another example, in the stock market, some low-frequency bandwidths may represent the value of the stock, while some high-frequency bandwidths may reflect noises that are not beneficial for investment. Although real-world data can be complex and noisy, making it challenging to strictly disentangle invariant and variant patterns, extensive experiments on real-world dynamic graphs empirically show that our method can better handle distribution shifts on dynamic graphs than the baselines. We will add the discussion in the revised paper.
>
> > The proposed model is heavily based on the invariant learning framework on graph data, and the originality concerning the distribution shifts seems limited. The novel part from the frequency modeling can be an orthogonal technique that is also applicable for other problems beyond the out-of-distribution generalization, and the authors fail to argue the advantage and uniqueness of the frequency modeling w.r.t. the temporal OOD problem.
>
> We appreciate your perspective on the proposed model. To the best of our knowledge, this paper is the first to study distribution shifts on dynamic graphs in the spectral domain, and in section 3, we give a motivation example with theoretical analysis to bridge the relationship between the invariant/variant patterns in the spectral domain and OOD generalization. This frequency modeling on dynamic graph OOD generalization is unique, providing another spectral analytical perspective besides the temporal view in the literature, and tells us the advantage of achieving dynamic graph OOD generalization in the spectral domain. Based on these analyses, we design a DyGNN with Fourier transform to obtain the node spectrums, a disentangled spectrum mask to obtain invariant and variant spectrum masks in the spectral domain, and propose the invariant spectral filtering mechanism to handle distribution shifts. Extensive experiments also empirically verify the effectiveness of our method. Nevertheless, we agree that our method can be extended to other problems beyond OOD generalization in future works.
>
> > More comparisons with some static baselines, e.g., GCN, GraphSAGE, EERM and DIR.
>
> Thank you for your suggestion. Following your suggestion, we provide comparisons with more static baselines on the dynamic link prediction datasets collab and yelp. We merge the graph snapshots to a static graph as inputs to the baselines. The results are shown in the following table, where OOM denotes out-of-memory on NVIDIA 3090 with 24GB GPU memory.
>
> | Model | Collab | Yelp |
> |:---:|:---:|:---:|
> | GCN | 76.17+-0.31 | 65.43+-1.07 |
> | GraphSAGE | 59.93+-0.09 | 62.61+-0.08 |
> | GAT | 69.43+-0.26 | 61.66+-0.03 |
> | GCRN | 69.72+-0.45 |  54.68+-7.59 |
> | EGCN |  76.15+-0.91 |  53.82+-2.06 |
> | DySAT |  76.59+-0.20 |  66.09+-1.42 |
> | IRM |  75.42+-0.87 |  56.02+-16.08 |
> | VREx |  76.24+-0.77 |  66.41+-1.87 |
> | GroupDRO |  76.33+-0.29 |  66.97+-0.61 |
> | DIR | 77.01+-0.16 | 67.21+-0.52 |
> | EERM | OOM | 65.55+-2.13 |
> | DIDA |   81.87+-0.40 |   75.92+-0.90 |
> | Ours |   84.09+-0.16 |   78.65+-2.22 |
>
> As shown in the table above, although EERM and DIR have shown competitive performance in static graph OOD settings, they have limited improvements over static GNNs in dynamic link prediction, which may be due to their ignorance of tackling the time information that is essential in dynamic graphs. In contrast, our method has significant performance improvements over the baselines, which verifies the effectiveness of our method in handling distribution shifts on dynamic graphs. We appreciate your suggestion and will include the additional comparisons in the revised paper.

---

> > ### Comment · Reviewer_QjfT · 2023-08-15
> >
> > Thank the authors for clarifying these points. However, in light of the response, this paper still has some issues at the current stage.
> >
> > > *"We collect the variant patterns of all nodes to construct the variant pattern set and uniformly sample the variant patterns from the set without replacement."*
> >
> > My major concern stems from the rationale of the method. The authors assume the dynamic graph data is comprised of invariant and variant patterns which are unobserved, and the proposed model learns to disentangle these two patterns (Z_V and Z_I) in an unsupervised way. The disentangled Z_V and Z_I are then used for prediction and the model is trained with the supervised loss. Then the problems arise:
> >
> > - How can you guarantee the Z_V and Z_I estimated by the model are indeed the invariant and variant patterns, without any inductive bias or prompt for the model (notice that your NNs for predicting Z_V and Z_I are in the same form)
> >
> > - In Eqn 7, the prediction is purely based on either the invariant or variant patterns, instead of their combination, which can be problematic. For example, the variant patterns could also contain useful information for prediction, though it is sensitive to environments.
> >
> > - A follow-up issue lies in Eqn 9, the mixed loss is calculated by the Hadamard product between variant and invariant parts, which is inconsistent with the classifier in Eqn 7. So how can you guarantee Eqn 9 can have the desired regularization effect as the paper claims.
> >
> > The above issues lead to the inconsistency between the motivation from spectral perspective and what the model practically learns. As the authors' response turns out, it is questionable whether the heuristic designs from high-level intuitions can indeed implement the main ideas.
> >
> > > *"This frequency modeling on dynamic graph OOD generalization is unique, providing another spectral analytical perspective besides the temporal view in the literature, and tells us the advantage of achieving dynamic graph OOD generalization in the spectral domain."*
> >
> > Another concern is the limited novelty along with potential over-claiming regarding the method. As acknowledged by the authors, the proposed model is also applicable for other graph data and tasks beyond the OOD tasks on dynamic graphs. Notice that in broader graph learning and OOD generalization, there already exists invariant learning [1] and the disentangle-based model [2]. This resonates my impression that the present model is a direct adaptation of existing techniques to a narrow area, and the unique technical contributions are limited and not associated with the target area.
> >
> > > *"We merge the graph snapshots to a static graph as inputs to the baselines."*
> >
> > The response also suggests that the baselines are put in an unfair comparison with the proposed model. The proposed model uses a dynamic GNN that will take a sequence of graph snapshots as model inputs, while the baseline models use static GNN that only takes a single graph (by condensing the graph sequence into one graph) as input. In this way, it is natural to see the performance improvement since the baselines use incomplete data and less expressive GNNs. This also amplifies my concern on the validity of the experiments in Sec 5 where the common baselines for OOD tasks are all put in an disadvantageous comparison, and the question regarding the efficacy of the model still remains, i.e., whether the frequency invariant learning indeed contributes to the performance gain.

---

> > > ### Author Response · Authors · 2023-08-16
> > > **(1/3) Response to the Reviewer QjfT**
> > >
> > > We thank the reviewer for the active discussion. After carefully reading the reviewer's comments, we think there exist some misunderstandings and would like to clarify the concerns point by point.
> > >
> > > > *"How can you guarantee the Z_V and Z_I estimated by the model are indeed the invariant and variant patterns, without any inductive bias or prompt for the model (notice that your NNs for predicting Z_V and Z_I are in the same form)"*
> > >
> > > Thank you for your question. We would like to clarify that **the calculations of $\mathbf{Z}\_{V}$ and $\mathbf{Z}_{I}$ are not in the same form in our framework, and we explicitly introduce the inductive bias to encourage each module to capture the invariant and variant patterns respectively**.
> > >
> > > - First, $\mathbf{Z}\_{V}$ and $\mathbf{Z}\_{I}$ are **calculated with different masks**. The invariant and variant patterns are encouraged to be disentangled by the proposed disentangled mask mechanism. In Eq. (4), the disentangled masks calculated for invariant and variant patterns are $\sigma(\mathbf{M}/\tau)$ and $\sigma(-\mathbf{M}/\tau)$, respectively, so that the masks are complementary, and encourage $\mathbf{Z}\_{V}$ and $\mathbf{Z}\_{I}$ to be disentangled. This design also makes it impossible that the invariant and variant patterns to be learned to be the same.
> > >
> > > - Second, $\mathbf{Z}\_{V}$ and $\mathbf{Z}\_{I}$ are **trained with different supervised signals**. In Eq. (10), the task loss for invariant patterns, i.e., $\mathcal{L}\_I$, and the invariance loss, i.e., $\mathcal{L}_{INV}$, will only supervise and update the parameters except the classifier for variant patterns, which encourages the model to capture predictive invariant patterns, even under exposure to different variant patterns. On the other hand, the last term, $\mathcal{L}_V$, only updates the classifier for variant patterns. Therefore, the calculation and prediction of invariant and variant patterns are not the same or symmetric, and inductive bias of disentanglement and invariance is introduced in the optimization process.
> > >
> > > Furthermore, we would like to clarify that **we also introduce inductive bias by the proposed mechanism in our framework**.
> > >
> > > - First, in Eq. (10) the first term $\mathcal{L}_I$ updates the parameters of DyGNN, masks, and the classifiers for invariant patterns with the task loss, which is related to condition 1 in the assumption 1, encouraging the model to capture predictive invariant patterns.
> > >
> > > - Second, in Eq. (10) the second item $\mathcal{L}_{INV}$ measures the model's prediction abilities with exposure to various variant patterns, which is related to condition 3 in assumption 1, encouraging the model to rely on the invariant patterns to make predictions.
> > >
> > > These inductive bias introduced by the proposed mechanism and objectives encourage the $\mathbf{Z}_V$ and $\mathbf{Z}_I$ estimated by the model to capture the invariant and variant patterns.
> > >
> > > > *"In Eqn 7, the prediction is purely based on either the invariant or variant patterns, instead of their combination, which can be problematic. For example, the variant patterns could also contain useful information for prediction, though it is sensitive to environments."*
> > >
> > > Thank you for your comment. We would like to point out that the prediction is based ONLY on invariant, rather than variant patterns or the combination.
> > > **Since the information contained in variant patterns can be sensitive to environments as you mentioned, they can lead to the overfitting of environment-related information and act as shortcuts to let the model suffer from the distribution shifts**. Similar phenomena have been observed in the OOD literature [1-2].
> > >
> > > To generalize under distribution shifts, our framework aims to exploit the invariant patterns, which have sufficient predictivities even with exposure to variant patterns and also have invariant relationships between labels under distribution shifts, to make predictions. Experiments on node/link real-world/synthetic dynamic graph datasets also empirically verify the effectiveness of only exploiting invariant patterns to make predictions under distribution shifts. Moreover, Proposition 1 in the main paper shows that if the model relies on the variant patterns on dynamic graphs to make predictions, the model could have unbound errors under distribution shifts, making the model unreliable due to the variant patterns. To handle distribution shifts on dynamic graphs, we only adopt invariant patterns to make predictions to avoid the harm of variant patterns.
> > >
> > > [1] Arjovsky, Martin, et al. "Invariant risk minimization." arXiv preprint arXiv:1907.02893 (2019).
> > >
> > > [2] Krueger, David, et al. "Out-of-distribution generalization via risk extrapolation (rex)." International Conference on Machine Learning. PMLR, 2021.

---

> > > > ### Author Response · Authors · 2023-08-16
> > > > **(2/3) Response to the Reviewer QjfT**
> > > >
> > > > > *"A follow-up issue lies in Eqn 9, the mixed loss is calculated by the Hadamard product between variant and invariant parts, which is inconsistent with the classifier in Eqn 7. So how can you guarantee Eqn 9 can have the desired regularization effect as the paper claims."*
> > > >
> > > > Thank you for your question. We would like to clarify that **Eq. (9) is used in calculating the invariance loss in Eq. (8) , and Eq. (7) calculates the task losses, and they are distinct**. We also would like to clarify the rationales of Eq. (9). The goal of the mixed loss $\mathcal{L}_m | \mathbf{z}$ is to measure the model's predictive ability with exposure to the variant patterns. Inspired by [1], we adopt the implementation of Eq. (9) to calculate the prediction of invariant patterns with the presence of variant patterns by conducting the Haramard product between the invariant pattern class vectors and the sigmoid of the variant pattern class vectors. This implementation measures the model's prediction performance with a given variant pattern while the invariant patterns are kept unchanged. Therefore, the variance of the mixed losses, i.e., Eq. (8), acts as an invariance regularization term to encourage the model to focus on the invariant patterns to make predictions.
> > > >
> > > > [1] Cadene, Remi, et al. "Rubi: Reducing unimodal biases for visual question answering." Advances in neural information processing systems 32 (2019).
> > > >
> > > > > *"Another concern is the limited novelty along with potential over-claiming regarding the method. As acknowledged by the authors, the proposed model is also applicable for other graph data and tasks beyond the OOD tasks on dynamic graphs. Notice that in broader graph learning and OOD generalization, there already exists invariant learning [1] and the disentangle-based model [2]. This resonates my impression that the present model is a direct adaptation of existing techniques to a narrow area, and the unique technical contributions are limited and not associated with the target area."*
> > > >
> > > > Thank you for your comment. We would like to clarify that **our method has special theoretical analyses and model designs for dynamic graphs under distribution shifts, and we are the first to handle distribution shifts on dynamic graphs in the spectral domain, to the best of our knowledge**.
> > > >
> > > > - First, **we give a clear motivation example and theoretical analysis** to show how the dynamic graph model under distribution shifts fails in the time domain while having the potential of generalization in the spectral domain. This perspective of tackling dynamic graph distribution shifts in the spectral domain remains unexplored in the dynamic graph literature.
> > > >
> > > > - Second, it is non-trivial to capture different dynamic graph patterns driven by various frequency components entangled in the spectral domain and to handle distribution shifts on dynamic graphs. To tackle this problem, we propose a framework named spectral invariant learning for dynamic graphs under distribution shifts, which can handle distribution shifts on dynamic graphs in the spectral domain.
> > > >
> > > > - Third, **the modules in the proposed framework are specifically designed for handling distribution shifts on dynamic graphs in the spectral domain**. The proposed DyGNN with Fourier transform is designed to simultaneously consider the graph structures, temporal features, structural evolution as well as capturing the spectral patterns. The proposed disentangled spectrum mask obtains the spectrum masks for invariant and variant patterns on dynamic graphs, and the proposed invariant spectral filtering encourages the model to rely on invariant patterns to make predictions on dynamic graphs.
> > > >
> > > > - Fourth, **our method is not a direct adaptation of existing techniques**. Most existing invariant learning or disentangled-based methods only consider the distribution shifts of either structures or time series, without considering the spatio-temporal information on dynamic graphs, e.g., how the evolution patterns of structures and features affect the predictions. However, these spatio-temporal information are essential in dynamic graphs, and the spatio-temporal distribution shifts ubiquitously exist in real-world dynamic graph scenarios. In contrast, the proposed framework in this paper can simultaneously consider the graph structures, temporal features, structural evolution as well as spectral patterns, to handle the distribution shifts on dynamic graphs.

---

> > > > > ### Author Response · Authors · 2023-08-16
> > > > > **(3/3) Response to the Reviewer QjfT**
> > > > >
> > > > > > *"The response also suggests that the baselines are put in an unfair comparison with the proposed model. The proposed model uses a dynamic GNN that will take a sequence of graph snapshots as model inputs, while the baseline models use static GNN that only takes a single graph (by condensing the graph sequence into one graph) as input. In this way, it is natural to see the performance improvement since the baselines use incomplete data and less expressive GNNs. This also amplifies my concern on the validity of the experiments in Sec 5 where the common baselines for OOD tasks are all put in an disadvantageous comparison"*
> > > > >
> > > > > Thank you for the comment. We would like to clarify that **the comparisons between the baselines and the proposed model are fair** for the following three reasons:
> > > > >
> > > > > - First, **dynamic graph baselines are included in our comparisons**, and *it is a misunderstanding that only `less expressive GNNs'(static GNNs) are included in the experiments*. The dynamic graph baselines include GCRN, EGCN, DySAT, and DIDA in all experiments.
> > > > >
> > > > > - Second, **common OOD baselines adopt the best-performed dynamic GNN (DySAT) as their backbones rather than static GNNs**, and *it is a misunderstanding that the common baselines for OOD tasks are all put in a disadvantageous comparison*.
> > > > >
> > > > > - Lastly, **even for the static baselines, the training data is complete without any information loss**, and *it is a misunderstanding that the training data to baselines are incomplete*. Specifically, denote the time span for training data as $T$, and the graph at time $t$ as $\mathcal{G}_t$. The training input to all the dynamic GNN methods is $( \mathcal{G}_1, \mathcal{G}_2, \dots, \mathcal{G}_T)$, and the training input to the static GNN methods is $\mathcal{G} = \text{MERGE}( \mathcal{G}_1, \mathcal{G}_2, \dots, \mathcal{G}_T)$. Therefore, all the methods have the complete training data. Note that static GNNs can only take one graph as input, so this is the common practice in comparing with them, as in the previous works [1-2].
> > > > >
> > > > > [1] Sankar, Aravind, et al. "DySAT: Deep neural representation learning on dynamic graphs via self-attention networks." Proceedings of the 13th international conference on web search and data mining. 2020.
> > > > >
> > > > > [2] Skarding, Joakim, Bogdan Gabrys, and Katarzyna Musial. "Foundations and modeling of dynamic networks using dynamic graph neural networks: A survey." IEEE Access 9 (2021): 79143-79168.
> > > > >
> > > > > > *"The question regarding the efficacy of the model still remains, i.e., whether the frequency invariant learning indeed contributes to the performance gain."*
> > > > >
> > > > > Thank you for your comment. We would like to explain that **we have verified the effectiveness of the spectral invariant learning through the ablation studies**. Specifically, in the main paper Section 5.3, we conduct the ablation studies of the invariant spectral filtering and disentangled spectrum masks on all datasets, where the ablated version 'SILD w/o I' removes the invariant spectral filtering and 'SILD w/o I' removes the disentangled spectrum masks. Figure 3 shows that 'SILD w/o I' drop drastically in performance on all datasets compared to the full version, which verifies that the spectral invariant learning contributes to the performance gain.
> > > > >
> > > > > We will make sure to provide the necessary clarifications and avoid these misunderstandings in the revision. Should you have any further questions or concerns, we are glad to continue the discussion and provide further clarifications.

---

> > > > > > ### Comment · Reviewer_QjfT · 2023-08-18
> > > > > >
> > > > > > Thanks for the response. The motivation of variant/invariant decomposition is a well-established framework, which has been explored in general graph learning and used for dynamic graphs [1], so it is unnecessary to rephrase it again in the narrower context of spectral dynamic graphs. My main concern is towards the vagueness of the approach presented in this paper, whether the model using a straightforward mask can indeed learn the invariant frequency patterns. The author's response is limited to the intuition level and the present analysis is predicated on unrealistic simple cases. The connection between the frequency story and what the model practially learns seems pretty loose. Even for the experimental datasets, the appendix shows each graph dataset contains only a dozen snapshots, and it is impossible for the model to learn different frequency patterns from such a short sequence.
> > > > > >
> > > > > > After reading the baseline paper [1], it further amplifies my concern on the limited contributions of the paper. The present manuscript shares a large overlap with [1], and the contributions are not enough for a conference paper.
> > > > > >
> > > > > > [1] Dynamic Graph Neural Networks Under Spatio-Temporal Distribution Shift, NeurIPS22

---

> > > > > > > ### Author Response · Authors · 2023-08-18
> > > > > > >
> > > > > > > We thank the reviewer for the further feedback. We would like to provide responses to your newly mentioned questions as follows.
> > > > > > >
> > > > > > > > Each graph dataset contains only a dozen snapshots. Is it possible for the model to learn different frequency patterns?
> > > > > > >
> > > > > > > Thank you for your question. We would like to clarify that the number of snapshots only determines the basis for the frequency pattern, while different nodes actually provide different samples to learn the frequency patterns on a specific dataset. Since the number of nodes in Collab, Yelp, Link-Synthetic, Aminer, and Node-Synthetic is 23K, 13K, 23K, 43K, and 5K, respectively, there are at least thousands of samples, which makes it possible to learn different frequency patterns. Notice that spectral analyses have also been performed in sequential recommendation[1] and video captioning[2], where there also only exist a dozen snapshots, indicating the number of snapshots does not affect learning frequency patterns.
> > > > > > >
> > > > > > > [1] Du, Xinyu, et al. "Frequency Enhanced Hybrid Attention Network for Sequential Recommendation." Proceedings of the 46th International ACM SIGIR Conference on Research and Development in Information Retrieval. 2023.
> > > > > > >
> > > > > > > [2] Aafaq, Nayyer, et al. "Spatio-temporal dynamics and semantic attribute enriched visual encoding for video captioning." Proceedings of the IEEE/CVF conference on computer vision and pattern recognition. 2019.
> > > > > > >
> > > > > > > > The comparisons with DIDA [1].
> > > > > > >
> > > > > > > Thank you for your question. Though our paper and DIDA[1] both study distribution shifts on dynamic graphs, **the main idea and the model design of our method and DIDA are completely different**. Specifically, the main differences are provided as follows:
> > > > > > >
> > > > > > > - **Different research perspectives.** DIDA explores the distribution shifts on dynamic graphs in the time domain, while in this paper, our analyses show that there exist cases that patterns on dynamic graphs are indistinguishable in the time domain, but they can be clearly distinguished in the spectral domain.
> > > > > > > - **Different methodologies.** Based on these observations and our theoretical analyses, we propose a framework to tackle the distribution shifts on dynamic graphs in the spectral domain with specially designed modules, which simultaneously consider the evolving structures, features as well as the spectral patterns that are ignored in DIDA.
> > > > > > > - **Superior empirical performance.** Experiments show that our method consistently and significantly outperforms DIDA on real-world/synthetic node/link prediction datasets, and the ablation studies verify the effectiveness of the designed modules.
> > > > > > >
> > > > > > > Thank you again for the feedback. Please let us know if you have further comments.
> > > > > > >
> > > > > > > [1] Dynamic Graph Neural Networks Under Spatio-Temporal Distribution Shift, NeurIPS22

---

> > > > > > > > ### Comment · Reviewer_QjfT · 2023-08-19
> > > > > > > >
> > > > > > > > Thank you for the clarification. Then the paper still has some serious issues.
> > > > > > > >
> > > > > > > > "We would like to clarify that the number of snapshots only determines the basis for the frequency pattern, while different nodes actually provide different samples to learn the frequency patterns on a specific dataset."
> > > > > > > >
> > > > > > > > If this interpretation holds, then the motivation example is flawed. The long-term/short-term interests of users and the value/noise of the stock are based on frequency patterns in time domain. But the authors insert that the model actually learns frequency patterns from the graph. The sequential recommendation and video captioning are different problems where the observed sequence for each sample is different and independent. But in the studied case, there is only one sequence of graph snapshots. It is impossible to learn the complex invariant frequency patterns from only one graph sequence in the dataset. The inconsistent logic and vagueness of the method make the manuscript not yet ready for publication.

---

> > > > > > > > > ### Author Response · Authors · 2023-08-20
> > > > > > > > >
> > > > > > > > > We thank the reviewer for the further feedback. We would like to clarify that our paper is consistent. Since the reviewer seems to agree that the issue regarding length has been addressed, we focus on the “number of samples” part. As we explained in the last response, though there is only one sequence of graph snapshots, different nodes provide different samples to learn invariant frequency patterns. Compared to sequential recommendations and video captioning, the samples in dynamic graphs are indeed dependent, which is the unique challenge brought by graphs, but this does not collapse the data into one sample. For instance, different users can be considered as different samples with complicated interactions. We design tailored modules to capture these complex frequency patterns, including a DyGNN with Fourier Transform, disentangled spectrum mask and invariant spectral filtering. Note that we focus on node and edge-level tasks, so we only learn node representations rather than the whole graph representation, e.g., in the above case, we do not aim to learn a representation for the whole e-commerce network, where only one sample exists, as the reviewer suggested. We will further clarify the expression in the revision.
> > > > > > > > >
> > > > > > > > > Thank you again for the feedback. Please let us know if you have further comments.

---

> > > > > > > > > > ### Comment · Reviewer_QjfT · 2023-08-21
> > > > > > > > > >
> > > > > > > > > > Thank you for the reply. I think I have no misunderstanding on the current submission. The paper has the following main issues:
> > > > > > > > > >
> > > > > > > > > > 1. The novelty and contributions are limited. The proposed model has no essential difference to the NeurIPS22 work DIDA which has explored invariant learning on dynamic graphs.
> > > > > > > > > >
> > > > > > > > > > 2. The motivation examples are flawed, and the model is vague. How the model actually learns invariant/variant frequency patterns is unclear. The theoretical results are trivial and the reproducibility is zero.
> > > > > > > > > >
> > > > > > > > > > 3. The comparison and evaluation are limited. Some important OOD baselines are not compared or weakened in the experiments. The datasets are also small with a dozen graph snapshots, arousing a question whether the model can indeed learn complex invariant frequency patterns. There is no concrete verification for this critical point.
> > > > > > > > > >
> > > > > > > > > > 4. The model section seems like a combination of existing works, but unfortunately, citations are not properly provided. For example, the model in Sec 4 has large overlap with DIR, the theoretical motivation in Sec 3 is similar to EERM, the experiment in Sec 5 is nearly the same as DIDA.
> > > > > > > > > >
> > > > > > > > > > I thus remain a clear reject recommendation for this paper.

---

> > > > > > > > > > > ### Author Response · Authors · 2023-08-21
> > > > > > > > > > >
> > > > > > > > > > > We thank the reviewer for the further feedback. We would like to provide responses to your new comments as follows.
> > > > > > > > > > >
> > > > > > > > > > > > Q1. The differences with DIDA.
> > > > > > > > > > >
> > > > > > > > > > > A1. We have provided detailed comparisons between our method and DIDA in previous responses. In short, we have different research perspectives (spectral vs. temporal), different methodologies (modules specially designed for tackling distribution shifts on dynamic graphs in the spectral domain), and superior empirical performance. These all make our paper and DIDA highly distinct, and the novelty is agreed upon by all other reviewers.
> > > > > > > > > > >
> > > > > > > > > > > > Q2-1. How the model actually learns invariant/variant frequency patterns
> > > > > > > > > > >
> > > > > > > > > > > A2-1. We have provided detailed clarifications about the motivation examples and model design in previous responses, including both theoretical, conceptual, and empirical results.
> > > > > > > > > > >
> > > > > > > > > > > > Q2-2. Regarding reproducibility.
> > > > > > > > > > >
> > > > > > > > > > > A2-2. We have provided the algorithm pseudocodes as well as the details of training and evaluation protocols, model implementation, and hyper-parameters in the main paper and Appendix. Moreover, the datasets are all publicly available. We will release the source codes at publication time.
> > > > > > > > > > >
> > > > > > > > > > > > Q3-1. Some important OOD baselines are not compared or weakened in the experiments.
> > > > > > > > > > >
> > > > > > > > > > > A3-1. We have provided the experimental results for various OOD baselines (IRM, VREx, GraphDRO, DIDA) and further have added baselines (GCN, GraphSAGE, EERM, DIR) in the rebuttal, following your suggestions. We have also clarified that the comparisons with them are fair and the experimental setting follows the existing literature.
> > > > > > > > > > >
> > > > > > > > > > > > Q3-2. Whether the model can indeed learn complex invariant frequency patterns.
> > > > > > > > > > >
> > > > > > > > > > > A3-2. We have provided detailed discussions about learning the complex invariant frequency patterns in previous responses.
> > > > > > > > > > >
> > > > > > > > > > > > Q4. Citations are not properly provided. For example, the model in Sec 4 has large overlap with DIR, the theoretical motivation in Sec 3 is similar to EERM, the experiment in Sec 5 is nearly the same as DIDA.
> > > > > > > > > > >
> > > > > > > > > > > A4. We respectfully clarify that **we have clearly cited all these papers in the original paper**, detailed as follows.
> > > > > > > > > > >
> > > > > > > > > > > (Regarding DIR) First, we would like to clarify that our paper and DIR have significant differences, since DIR focus on static graphs and is completely unrelated to the spectral domain. The only similarity lies in Eqs. (9)(10), and we have clearly cited DIR in the original paper (P6, L192-193)：*"Inspired by [34, 15], we calculate the mixed loss as ..."*, where [15] is DIR.
> > > > > > > > > > >
> > > > > > > > > > > (Regarding EERM) Our analysis indeed draws inspiration from EERM and we have clearly cited it as follows (P3, L92-93): *"In the next section, inspired by [16], we give a motivation example ..."*, where [16] is EERM. Besides, our theoretical analyses are more complex since we consider dynamic graphs, while EERM focuses on static graphs.
> > > > > > > > > > >
> > > > > > > > > > > (Regarding DIDA) Our paper and DIDA tackle the same research problem (though we study the problem from completely different perspectives, see A1) so we follow DIDA for the experimental setting. We have clearly cited it in the paper (P7, L223; P8, L251):"*Following [7], we adopt the challenging inductive future link prediction task ...", "For link prediction datasets, we follow [7] to generate additional varying features..."*, where [7] is DIDA.
> > > > > > > > > > >
> > > > > > > > > > > Thank you again for the feedback.

---

### Official Review · Reviewer_4zF7 · 2023-07-07

**Soundness:** 3 good
**Presentation:** 3 good
**Contribution:** 3 good
**Rating:** 8
**Confidence:** 4

**Summary:**

Dynamic graph neural networks are still facing the difficulty in handling distribution shifts. This work discovers that in some cases, distribution shifts are unobservable in the time domain but not in the spectral domain. There are currently no studies to handle cases involving distribution shifts in the spectral domain. To solve these problems, the authors propose Spectral Invariant Learning for Dynamic Graphs under Distribution Shifts. Specifically, a DyGNN with Fourier transform is designed to obtain the ego-graph trajectory spectrums. Furthermore, a disentangled spectrum mask to filter graph dynamics from various frequency components is developed. Finally, invariant spectral filtering is proposed, which can encourage the model to rely on invariant patterns for generalization under distribution shifts. Experimental results on synthetic and real-world dynamic graph datasets verified that this method is superior for node classification and link prediction tasks under distribution shifts.


**Strengths:**

This paper studies distribution shifts on dynamic graphs in the spectral domain for the first time. The idea for tackling temporal distribution shifts in the spectral domain is natural. The theoretical anaylses strengthen the motivations. Experimental results on synthetic and real-world dynamic graph datasets verified that this method is superior for node classification and link prediction tasks under distribution shifts.


**Weaknesses:**

1) It would be great to include the F1 scores as metrics for comparisons of different methods in node classification tasks.
2) In Figure 2, the calculation flow of the framework is not very clear, and it would be great to highlight the order of each module.
3) Some notations should be more clear. For example, the notation '||' in Eq. 4  and $\odot$ in Eq. 5.
4) It would be great to include more related works about spectral methods.


**Questions:**

1) It would be great to include the F1 scores as metrics for comparisons of different methods in node classification tasks.
2) In Figure 2, the calculation flow of the framework is not very clear, and it would be great to highlight the order of each module.
3) Some notations should be more clear. For example, the notation '||' in Eq. 4  and $\odot$ in Eq. 5.
4) It would be great to include more related works about spectral methods.


**Limitations:**

The authors state that this paper mainly focuses on dynamic graphs in scenarios of discrete snapshots, and leave extending the method to continuous dynamic graphs for further explorations.

---

> ### Author Rebuttal · Authors · 2023-08-09
>
> We sincerely thank the reviewer for the valuable suggestions. We respond to the reviewer’s comments point by point as follows.
>
> > It would be great to include the F1 scores as metrics for comparisons of different methods in node classification tasks.
>
> Thank you for your suggestion. Following your suggestion, we report the Macro-F1 scores of different methods on the node classification dataset Aminer in the following table.
>
> | Model | Aminer-15 | Aminer-16 | Aminer-17 |
> |---|---|---|---|
> | GCRN | 36.89+-0.42 | 37.65+-0.88 | 35.30+-0.78 |
> | EGCN | 33.47+-4.78 | 30.54+-3.77 | 28.02+-5.68 |
> | DySAT | 38.37+-1.27 | 38.19+-0.45 | 35.52+-1.02 |
> | IRM | 37.44+-0.82 | 37.28+-0.48 | 35.23+-1.24 |
> | VREx | 37.25+-0.58 | 37.11+-0.49 | 35.03+-0.54 |
> | GroupDRO | 38.30+-0.86 | 37.64+-1.03 | 35.44+-0.74 |
> | DIDA | 38.26+-0.37 | 38.45+-0.52 | 36.63+-1.08 |
> | Ours | 40.81+-0.41 | 40.17+-0.49 | 37.16+-0.96 |
>
> As shown in the table above, our method has a significant performance improvement over the baselines in terms of the metric of Macro-F1, which further verifies the effectiveness of our method in handling distribution shifts on dynamic graphs.
>
> > In Figure 2, the calculation flow of the framework is not very clear, and it would be great to highlight the order of each module. Some notations should be more clear. For example, the notation '||' in Eq. 4 and in Eq. 5.
>
> Thank you for your suggestions. We provide the calculation flow of the framework as follows: 1) First, given a dynamic graph, the dynamic graph neural networks with spectral transform obtain the ego-graph trajectory spectrums in the spectral domain. 2) Second, the disentangled spectrum mask leverages the amplitude and phase information of the spectrums to obtain invariant and variant spectrum masks.  3) Third, the invariant spectral filtering discovers the invariant and variant patterns via the disentangled spectrum masks and helps the model rely on the invariant patterns to make predictions under distribution shifts. In the revised version of the main paper, we will enhance the diagram to highlight the order of each module in the framework, providing a clearer visualization of the calculation flow. Additionally, we will make the notations more explicit and improve the clarity of the notation '||' in Eq. 4 and Eq. 5, clearly indicating that it represents the concatenation of the embeddings. We appreciate your feedback and will address these issues in the updated version.
>
> > It would be great to include more related works about spectral methods.
>
> Thank you for your suggestion. While we have covered spectral methods relevant to handling graph and time-series data, we acknowledge the importance of including more related works on spectral methods in the revised version of the main paper. We will incorporate recent works that focus on Fourier neural operators to introduce spectral analysis in downstream tasks, such as partial differential equations. Specifically, we will include references to works like [1], which proposes a multiwavelet-based method for compressing operator kernels, and [2], which introduces Fourier-enhanced blocks and Wavelet-enhanced blocks for capturing important time-series features in the frequency domain. We appreciate your suggestion and will provide a more comprehensive discussion of related spectral methods in the updated version of the main paper.
>
> [1] Gupta, Gaurav, Xiongye Xiao, and Paul Bogdan. "Multiwavelet-based operator learning for differential equations." Advances in neural information processing systems 34 (2021): 24048-24062.
>
> [2] Zhang, Xiang, et al. "Self-supervised contrastive pre-training for time series via time-frequency consistency." Advances in Neural Information Processing Systems 35 (2022): 3988-4003.

---

> > ### Comment · Reviewer_4zF7 · 2023-08-18
> >
> > Thank you for the detailed response with clarifications and additional evaluations. The issues raised by me have been addressed sufficiently. I believe the paper is of high quality that solves a valuable dynamic graph ood problem with a novel method, and the evaluations are convincing. Thus, I would like to raise my score from 7 to 8.

---

### Official Review · Reviewer_fiaV · 2023-07-08

**Soundness:** 3 good
**Presentation:** 3 good
**Contribution:** 3 good
**Rating:** 7
**Confidence:** 4

**Summary:**

This paper addresses the problem of dynamic graph learning under distribution shifts. Distinct from the previous work that focuses on the time domain, the authors argue that there exists patterns that are not easily distinguished in the time domain while could be easier to recognize in the spectral domain. Some theoretical analyses and intuitive examples have been provided to clearly present the motivations. Based on these analyses, the authors propose a method named spectral invariant learning for dynamic graphs under distribution shift. The extensive experiments show that the method can well tackle the distribution shifts on dynamic graphs, and the performance is worthy of recognition. Detailed ablation studies also verify the effectiveness of each proposed components.

**Strengths:**

- This work tackles the distribution shifts on dynamic graphs in the spectral domain for the first time. The paper is well written and easy to follow.

- Some theoretical analyses and intuitive examples have been provided to clearly present the motivations. The motivations and the method designs sound reasonable.

- The extensive experiments show that the method can well tackle the distribution shifts on dynamic graphs, and the performance is worthy of recognition.

**Weaknesses:**

Please see the Questions.

**Questions:**

- In line 87, the condition 2 assumes that the observed data is composed of invariant and variant patterns. Is it possible that there exist multiple invariant and variant patterns?

- Some distribution shifts may stem from selection bias, e.g., some users in social networks may have much fewer or even no interactions in some periods. These issue let some temporal signals miss in some periods and nodes. And this seems to cause difficulties for the subsequent spectral analysis. It would be better to discuss this problem in the main paper.

- The paper adopts Fourier transform as an implementation of spectral transform. It would be better to discuss whether other spectral transforms can be adopted.

**Limitations:**

None.

---

> ### Author Rebuttal · Authors · 2023-08-09
>
> We would like to express our sincere appreciation to the reviewer for providing us with detailed comments and suggestions. We have carefully reviewed each comment and offer the following responses:
>
> > In line 87, the condition 2 assumes that the observed data is composed of invariant and variant patterns. Is it possible that there exist multiple invariant and variant patterns?
>
> Thank you for raising this point. We acknowledge that there can indeed be multiple invariant and variant patterns present in the observed data. In this paper, we define invariant patterns as those that maintain invariant relationships with the labels under distribution shifts, while variant patterns encompass all patterns that are not defined as invariant. Our disentangled spectrum mask is designed to handle cases with multiple invariant patterns by assigning higher mask scores to these patterns. Empirical results from our experiments on real-world datasets have demonstrated the effectiveness of our method in addressing complex scenarios involving real-world dynamic graphs under distribution shifts.
>
> > Some distribution shifts may stem from selection bias, e.g., some users in social networks may have much fewer or even no interactions in some periods. These issue let some temporal signals miss in some periods and nodes. And this seems to cause difficulties for the subsequent spectral analysis. It would be better to discuss this problem in the main paper.
>
> Thank you for bringing up this important point. We acknowledge that distribution shifts originating from selection bias can indeed result in missing temporal signals and nodes in certain periods. To address this challenge, our model leverages the past graph trajectories to obtain node embeddings for nodes that may lack interactions during certain time intervals. Specifically, our model incorporates the past interactions in addition to the current interactions when aggregating neighborhood information and updating node embeddings. This approach helps mitigate the impact of missing signals to some extent. We have provided more detailed information about the model in Appendix D.1 to elaborate on this aspect.
>
> > The paper adopts Fourier transform as an implementation of spectral transform. It would be better to discuss whether other spectral transforms can be adopted.
>
> We appreciate your suggestion. In this paper, our primary focus is on handling distribution shifts on dynamic graphs in the spectral domain, which remains an underexplored area in the literature. We have chosen Fourier transform due to its efficiency, semantic interpretability, and ability to capture global and periodic patterns [1-2]. We will extend our method to incorporate other spectral transforms such as Wavelet transform or more sophisticated techniques in future research.
>
> [1] Brigham, E. Oran. The fast Fourier transform and its applications. Prentice-Hall, Inc., 1988.
>
> [2] Yi, Kun, et al. "Neural Time Series Analysis with Fourier Transform: A Survey." arXiv preprint arXiv:2302.02173 (2023).

---

> > ### Comment · Reviewer_fiaV · 2023-08-18
> > **Score raise to 7**
> >
> > I appreciate the authors' response, which is both clear and effectively addresses my concerns. I have also reviewed the responses provided by the other reviewers, and from my side, I find the paper to be novel and well-founded overall. In particular, the spectral part for dynamic graphs is particularly inspiring to me.
> >
> > Therefore, I have raised my score to 7 for a support.

---

### Official Review · Reviewer_vwaL · 2023-07-08

**Soundness:** 3 good
**Presentation:** 3 good
**Contribution:** 3 good
**Rating:** 8
**Confidence:** 4

**Summary:**

This paper finds that there are cases with distribution shifts observable in the spectral domain while difficult to distinguish in the time domain. Based on this characteristic, the authors propose  Spectral Invariant Learning for Dynamic Graphs under Distribution Shifts (SILD) to handle distribution shifts on dynamic graphs in the spectral domain. A DyGNN with Fourier transform is designed to obtain the ego-graph trajectory spectrums. In addition, it develops a disentangled spectrum mask to filter graph dynamics and invariant spectral filtering to encourages the model to rely on invariant patterns. Experimental results verifies the superiority of this method under distribution shifts.

**Strengths:**

1. Distribution shifts naturally exist in real-world dynamic graphs. Handling these issues is critical for the reliable applications of dynamic GNNs. The problem studied in this paper is valuable and the research perspective is novel.

2. The paper is well-organized, and the analyses and examples given in the manuscript are easy to understand.

3. Experimental results verify the superiority of this method under distribution shifts.

**Weaknesses:**

1. In Figure 2, the spectrums block (bottom-middle) seems a little confusing. I guess it shows a temporal signal of a node can be decomposed into amplitudes and phases in the spectral domain.

2. In Line 195, what does the parameters $\theta$ contain specifically?

3. Typo in Table 2 caption.

4. The construction of synthetic datasets seems complex. Can you provide the intuitions concisely?

5. It seems $\epsilon$ refers to random noise in Assumption 1, while it represents an arbitrary number in the proof.

**Questions:**

Check the above weaknesses.

---

> ### Author Rebuttal · Authors · 2023-08-09
>
> We would like to express our sincere gratitude to the reviewer for providing us with detailed comments and insightful questions. We have carefully considered the reviewer's feedback and would like to address each point as follows:
>
> > In Figure 2, the spectrums block (bottom-middle) seems a little confusing. I guess it shows a temporal signal of a node can be decomposed into amplitudes and phases in the spectral domain.
>
> Thank you for your comment. Indeed, the spectrums block in Figure 2 illustrates the decomposition of a temporal signal of a node into amplitudes and phases in the spectral domain. To achieve this, we utilize dynamic graph neural networks with spectral transform to obtain ego-graph trajectory spectrums. Subsequently, the disentangled spectrum mask makes use of the amplitude and phase information from these spectrums to derive invariant and variant spectrum masks. We acknowledge the confusion caused by the current depiction and will make improvements to the diagram in the revised version of the main paper.
>
> > In Line 195, what does the parameters contain specifically?
>
> Thank you for bringing this to our attention. The parameters denoted as $\theta$ encompass all the network parameters, excluding the classifiers. These parameters are primarily associated with the dynamic graph neural network and disentangled spectrum masks. In response to your feedback, we will provide more detailed descriptions of the parameters in the revised version of the main paper.
>
> > Typo in Table 2 caption. It seems $\epsilon$ refers to random noise in Assumption 1, while it represents an arbitrary number in the proof.
>
> We appreciate your keen observation. You are correct that there is an inconsistency in the use of the symbol $\epsilon$. In Assumption 1, $\epsilon$ indeed refers to random noise, while in the proof of the propositions, $\epsilon$ represents an arbitrary positive real number. We apologize for this oversight and will rectify the typo in the caption. Additionally, we will ensure consistency in the notation throughout the revised version of the main paper.
>
> > The construction of synthetic datasets seems complex. Can you provide the intuitions concisely?
>
> We are grateful for your feedback. To provide a more concise and intuitive explanation of the construction of synthetic datasets, we offer the following summary. The underlying idea is to introduce spurious correlations in certain features and structures, requiring the model to rely on invariant patterns rather than variant ones when making predictions, even in the presence of distribution shifts.
>
> For link synthetic datasets, we generate additional varying features that are concatenated with the original features. These additional features are specifically designed to exhibit spurious correlations with the labels, which represent the links at the next time step.
>
> For node synthetic datasets, we generate dynamic graphs using a stochastic block model, where the probability of a link between nodes is determined by two frequency factors. One of these factors exhibits spurious correlations with the labels. The extent of spurious correlations in both datasets can be controlled using a parameter.
>
> We appreciate your feedback and will revise the description to provide a clearer and more concise explanation in the updated version of the paper.

---

> > ### Comment · Reviewer_vwaL · 2023-08-19
> > **Thanks for your reply.**
> >
> > My concerns have been addressed.

---

### Decision · Program_Chairs · 2023-09-21

**Decision:**

Accept (poster)

**Comment:**

This paper provides a new algorithm to learn dynamic graph neural networks which decomposes features into variant and invariant ones on the *frequency* domain. By doing so, it is claimed that the proposed model can outperform existing methods especially in OOD settings.
The proposed idea is very simple extension from the exiting DIDA to a frequency domain while DIDA deals with the time domain. The proposed method is supported by some theoretical explanation and numerical experiments. In particular, the numerical experiments show effectiveness of the proposed method.
Although there are some confusing description and the novelty of the method is rather limited given DIDA, I personally think this paper provides a practically useful method in the line of research. I hence recommend acceptance.

However, some notations are not clearly defined and learning procedure can be described in a clearer way. I strongly encourage the author to fix them.